# Data augmentation for efficient learning from parametric experts

**Alexandre Galashov**
DeepMind
agalashov@deepmind.com

**Josh Merel**[*]
DeepMind
jsmerel@gmail.com

**Nicolas Hess**
DeepMind
heess@deepmind.com

## Abstract

We present a simple, yet effective data-augmentation technique to enable data-efficient learning from parametric experts for reinforcement and imitation learning. We focus on what we call the *policy cloning* setting, in which we use online or offline queries of an expert or expert policy to inform the behavior of a student policy. This setting arises naturally in a number of problems, for instance as variants of behavior cloning, or as a component of other algorithms such as DAGGER, policy distillation or KL-regularized RL. Our approach, *augmented policy cloning* (APC), uses synthetic states to induce feedback-sensitivity in a region around sampled trajectories, thus dramatically reducing the environment interactions required for successful cloning of the expert. We achieve highly data-efficient transfer of behavior from an expert to a student policy for high-degrees-of-freedom control problems. We demonstrate the benefit of our method in the context of several existing and widely used algorithms that include policy cloning as a constituent part. Moreover, we highlight the benefits of our approach in two practically relevant settings (a) *expert compression*, i.e. transfer to a student with fewer parameters; and (b) transfer from *privileged experts*, i.e. where the expert has a different observation space than the student, usually including access to privileged information.

## 1 Introduction

In various control and reinforcement learning settings, there is a need to transfer behavior from an expert policy to a student policy. Broadly, when only samples from the expert policy are available, the standard approach is to employ a version of regression from states to actions. This class of approaches for producing a policy is known as behavioral cloning [Pomerleau, 1989, Michie and Sammut, 1996]. Behavioral cloning is quite flexible and supports the setting where the expert trajectories come from a human teleoperating the relevant system directly, as well as various settings where the trajectories are sampled from other controllers, which themselves may have been trained or scripted. However, for any of the settings where the expert policy is actually available, rather than just samples from the expert, it is reasonable to suspect that sampling random rollouts from the expert policy followed by performing behavioral cloning is not the most efficient approach for transferring behavior from the expert to the student. Once a trajectory has been sampled via an expert rollout, there is actually additional information available that can be ascertained in the neighborhood of the trajectory, without having to perform an additional rollout, via the local feedback properties of the expert.

We refer to this setting, where we want to transfer from an expert policy to a student policy, while assuming the expert policy can be queried, as *policy cloning*. Naturally, there is still often an incentive to reduce the total number of rollouts, which may require actually collecting data in an unsafe or costly fashion, especially for real-world control problems. As such, there is a motivation to characterize any

---

[*]Work was done while at DeepMind

36th Conference on Neural Information Processing Systems (NeurIPS 2022).

efficiency that can be gained in learning from small numbers of rollouts without as much concern for how many offline queries are required of the expert policy.

If one has primarily encountered behavioral cloning in the context of learning from human demonstrations, policy cloning, with an available expert policy may seem contrived. However, policy cloning naturally arises in many settings. For example, an expert policy may be too large to execute due to memory considerations as in Parisotto and Salakhutdinov [2021], where authors propose to distill a large transformer network into small MLPs to be able to execute the policy on data collection workers. In a different setting, we could aim to distill an expert which has access to additional privileged information into a student without access to it, for example by distilling a full state policy into the one which has vision observations. In applications such as robotics control tasks where observation are provided via vision, it is typically much easier to train a policy if some additional features are provided to the observations. However, these features might not be available when the policy should be used on the real robot. And in the DAGGER setting [Ross et al., 2011], a student policy collects data and is trained by regressing on the expert policy where state distribution comes from the student. In yet another setting, the expert may be suboptimal and the student needs to learn from expert while also being able to exceed the expert performance, perhaps by continuing to learn from a task via RL. This problem has been described as *kickstarting* in one incarnation [Schmitt et al., 2018], but also can arise when learning from behavioral priors [Tirumala et al., 2020], [Galashov et al., 2019], as also happens, for example, in Distral [Teh et al., 2017]. In all of the aforementioned situations, we implicitly rely on the assumption that it is cheap to query the expert policy. This, however, is not typically a restriction and the main motivation is to minimize the amount of data a student policy needs to collect.

To improve data-efficiency in supervised settings generally, including in behavioral cloning settings, it is reasonable to consider data augmentation. Data augmentation refers to applying perturbations to a finite training dataset to effectively amplify its diversity, usually in the hopes of producing a model that is invariant to the class of perturbations performed. For example, in the well studied problem of object classification from single images, it is known that applying many kinds of perturbation should not affect the object label, so a model can be trained with many input perturbations all yielding the same output [Shorten and Khoshgoftaar, 2019]. This setting is fairly representative, with data augmentation usually intended to make the model "robust" to nuisance perturbations of the input. This class of image-perturbation has also been recently demonstrated to be effective in the context of control problems in the offline RL setting [Yarats et al., 2021, Laskin et al., 2020].

Critically, for control problems it is not the case that the action should be invariant to the input state. Or rather, while it does make sense for a control policy to be invariant to certain classes of sensor noise, an important class of robustness is that the policy is appropriately feedback-responsive. This is to say that for small perturbations of the state of the control system, the optimal action is different in precisely the way that the expert implicitly knows. This has been recognized and exploited in previous research that has distilled feedback-control plans into controllers [Mordatch and Todorov, 2014, Mordatch et al., 2015, Merel et al., 2019]. A similar intuition also underlies schemes which inject noise into the expert during rollouts to sample more comprehensively the space of how the expert recovers from perturbations [Laskey et al., 2017, Merel et al., 2019]. Moreover, the idea of recovery from perturbations was explored in scenarios where strong domain knowledge is avaialable. In [Bojarski et al., 2016] for imitation learning for self-driging cars, the authros propose to shift the camera images and associated shifted labels with these, which requires to know the model of the shift. In [George et al., 2018], the authors used the label augmentation approach by assigning fixed label to syntetically created images. In [Buhet et al., 2019], the authors proposed to perturb the inputs and associate the labels which corresponded to the recovery of the perturbed inputs. Even if these approaches are related to our work, they require a strong domain knowledge and in some cases the knowledge of the perturbation model. In our approach presented below, we do not require any domain knowledge, but only the ability to query the expert policy.

In this work, we leverage this insight to develop a highly efficient policy cloning approach that makes use of both classes of data augmentation. For a high-DoF control problem that operates only from state (humanoid run and insert peg tasks from DeepMind control suite [Tunyasuvunakool et al., 2020]), we demonstrate the feasibility of policy cloning that employs state-based data augmentation with expert querying to transfer the feedback-sensitive behavior of the expert in a region around a small number of rollouts. Then on a more difficult high-DoF control problem that involves both state-derived and egocentric image observations (humanoid running through corridors task from DeepMind

control [Tunyasuvunakool et al., 2020]), we combine the state-based expert-aware data augmentation with a separate image augmentation intended to induce invariance to image perturbations. Essentially our expert-aware data augmentation involves applying random perturbations to the state-derived observations, and training the student to match the expert-queried optimal action at each perturbed state, thereby gaining considerable knowledge from the expert without performing excessive rollouts simply to cover the state space around existing trajectories. Our approach compares favorably to sensible baselines, including the naive approach of attempting to perform behavioral cloning with state perturbations, which seeks to induce invariance (as proposed in Laskin et al., 2020) rather than feedback-sensitivity to state-derived observations. We demonstrate that our approach significantly improves data efficiency on all the settings mentioned above, i.e., behavioral cloning, *expert compression*, cloning *privileged experts*, *DAgger* and *kickstarting*.

## 2 Problem description

### 2.1 Expert-driven learning

We start by introducing a notion of expert-driven learning that will be used throughout the paper. At first, we present a general form of the expert-driven objective and then introduce a few concrete examples. We consider a standard Reinforcement Learning (RL) problem. We present the domain as an MDP with continuous states for simplicity, however the problem definition is similar for a POMDP with observations derived from the state. Formally, we describe the MDP in terms of a continuous state space $\mathcal{S} \in \mathcal{R}^n$, $n > 0$, an action space $\mathcal{A}$, transition dynamics $p(s'|s,a) : \mathcal{S} \times \mathcal{A} \to p(\mathcal{S})$, and a reward function $r : \mathcal{S} \times \mathcal{A} \to \mathcal{R}$. Let $\Pi$ be a set of parametric policies, i.e. of mappings $\pi_\theta : \mathcal{S} \to p(\mathcal{A})$ from the state space $\mathcal{S}$ to the probability distributions over actions $\mathcal{A}$, where $\theta \in \mathcal{R}^m$ for some $m > 0$. For simplicity of the notation, we omit the parameter in front of the policy, i.e. $\pi = \pi_\theta$ and optimizing over the set of policies would be equivalent to the optimizing over a set of parameters. A reinforcement learning problem consists in finding such a policy $\pi$ that it maximizes the expected discounted future reward:

$$J(\pi) = \mathbb{E}_{p(\tau)} \left[ \sum_t \gamma^t r(a_t|s_t) \right], \tag{1}$$

where $p(\tau) = p(s_0) \prod_t p(a_t|s_t) p(s_{t+1}|s_t, a_t)$ is a trajectory distribution. We assume the existence of an expert policy $\pi_E(a|s)$. This policy could be used to simplify the learning of a new policy on the same problem. Formally, we construct a new learning objective which aims to maximize the expected reward of the problem at hand as well as to clone the expert policy:

$$J(\pi, \pi_E) = \alpha J(\pi) - \lambda D(\pi, \pi_E), \tag{2}$$

where $D$ is a function which measures the closeness of $\pi$ to $\pi_E$ and $\alpha \geq 0, \lambda \geq 0$ are parameters describing importance of both objectives. In most of the applications, $\alpha \in \{0, 1\}$ and $\lambda \geq 0$ represents a relative importance of cloning an expert policy with respect to the RL objective.

### 2.2 Behavioral cloning (BC)

Behavioral cloning (BC) corresponds to optimizing the objective (2) with $\alpha = 0, \lambda = 1$ where $D$ is:

$$D_{BC}(\pi, \pi_E) = -\mathbb{E}_{(a,s) \in \mathcal{B}_E} [\log \pi(a|s)] \tag{3}$$

Here, $\mathcal{B}_E = \{(s_i, a_i), i = 1, \ldots, N\}$, $N > 0$ is a fixed dataset containing expert data. Minimizing the objective (3) is equivalent to maximizing the likelihood of the expert data under the policy $\pi$. The action $a$ in eqn. (3) can be replaced by $\pi_E(s)$ for deterministic policies or by the mean $\mu_E(s)$ for Gaussian policies $\pi_E(\cdot|s) = \mathcal{N}(\mu_E(s), \sigma_E(s))$.

### 2.3 Expert compression

In case of expert compression, we are interested in optimizing a similar objective as in eqn. (3), but where the student $\pi$ has smaller number of parameters compared to $\pi_E$.

## 2.4 Learning from privileged experts

In case of learning from privileged experts, we optimize similar objective to eqn. (3), where student receives different observations from the expert $\pi_E$. We assume that the expert has access to the privileged information, but the student does not. In particular, we consider the case where dataset with expert data contains observations (rather than full state), i.e. $\mathcal{B}_E = \{(o_i, a_i), i = 1, \ldots, N\}$, $N > 0$, but the expert has access to the full state $s_i$. More precisely, we consider an expert that observes the state $s = (s_c, s_{priv})$, where $s_c$ is a set of observations common to both the student and expert, whereas $s_{priv}$ is privileged information (containing some task-specific information). Then, the observations for the student are obtained as $o = (s_c, o_{vis})$ where $o_{vis}$ is the vision-based input.

## 2.5 DAGGER

Performance of Behavioral Cloning (BC) can be limited due to the fixed dataset, since the resulting policy may fail to generalize to states outside the training distribution. A different approach, known in the literature as DAGGER [Ross et al., 2011] was proposed to overcome this limitation. In this setting, the expert is queried in states visited by the student, thus reducing distribution shift. In our notation, this corresponds to $\alpha = 0$, $\lambda = 1$ in eqn. (2) and D is defined as:

$$D_{\text{DAGGER}}(\pi, \pi_E) = -\mathbb{E}_{p_\beta(\tau)}[\log \pi(a_t' | s_t)], \tag{4}$$

where $p_\beta(\tau)$, $\beta \in [0, 1]$ is a trajectory distribution where actions are sampled according to the mixture policy between a student and an expert:

$$p_\beta(a|s) = \beta \tilde{\pi}(a|s) + (1 - \beta)\pi_E(a|s), \tag{5}$$

The action $a_t'$ in eqn. (4) is obtained from the expert policy as $a_t' \sim \pi_E(\cdot | s_t)$, as $\pi_E(s)$ for deterministic or as $\mu_E(s)$ for Gaussian policies $\pi_E(\cdot|s) = \mathcal{N}(\mu_E(s), \sigma_E(s))$ (see Section 2.2). The policy $\tilde{\pi}(a|s)$ corresponds to a frozen version of student policy $\pi$ so that the gradient $\nabla_\pi D_{\text{DAGGER}}(\pi, \pi_E)$ ignores the acting distribution $p_\beta(a|s)$. Note that even though, in eqn. (4) we collect data from the environment, the setting nevertheless corresponds to pure imitation learning.

## 2.6 Kickstarting

In eqn. (2), we combine both maximization of expected task reward and minimization of distance to the expert. In literature, it is known as *Kickstarting* [Schmitt et al., 2018]. In this case, the objective from eqn. (2) becomes:

$$J(\pi, \pi_E) = J(\pi) - \lambda \mathbb{E}_{p(\tau)} \left[ -\mathbb{E}_{\pi_E(a|s)} \log \pi(a|s) \right] \tag{6}$$

where $p(\tau)$ is a trajectory distribution, where actions are sampled according to the student policy $\pi(\cdot|s)$. It corresponds to having $\alpha = 1$, and $\lambda \geq 0$ and $D$ be state-conditional (across trajectory) cross-entropy from expert to a student. Usually, in the *Kickstarting* setting, the expert is sub-optimal and the goal is to train a policy that eventually outperforms the expert. Thus, it is customary to reduce $\lambda$ over the course of training. Yet, for simplicity, in our experiments we keep this coefficient fixed.

## 3 Augmented policy cloning

The previous section has demonstrated how the goal of cloning expert behavior can arise in different scenarios. In this section we propose a new and simple method which can significantly improve the data efficiency in the settings described in Section 2. We explain the basic idea for BC, but its generalization to other expert-driven learning approaches described in Section 2 is straightforward. In Section 6 we show results for these problems.

When optimizing the objective (3), for every state $s \in \mathcal{D}_E$ from the expert trajectories dataset, we consider a small Gaussian state perturbation:

$$\delta s \sim \mathcal{N}(0, \sigma_s^2) \tag{7}$$

which produces a new virtual state:

$$s' = s + \delta s \tag{8}$$

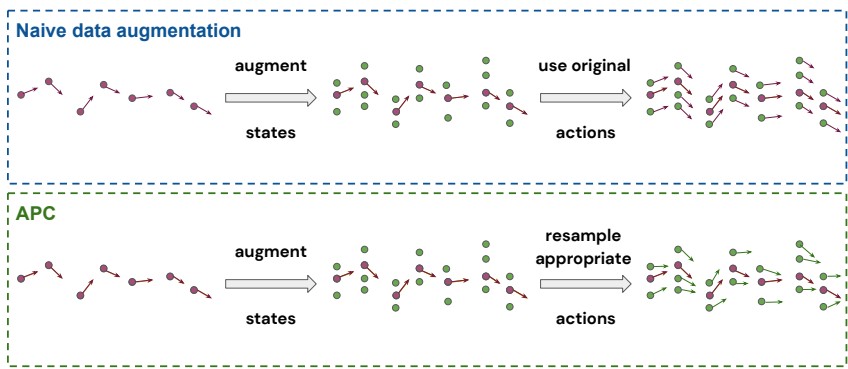

Figure 1: **Schematic of APC** (ours) method compared with a naive data augmentation approach. The original states (magenta circles) and actions (magenta arrows) pairs are then augmented by new virtual states (green circles). In Naive data augmentation, the same actions (magenta arrows) are used for all new virtual states. In APC, however, for each new virtual state, we resample a new action (green arrow) from the expert policy.

Then, for this state we query the expert and obtain a new action

$$a' \sim \pi_E(\cdot|s + \delta s) \tag{9}$$

We then augment the dataset $\mathcal{B}_E$ with these new pairs of virtual states and actions. More explicitly the idea can be expressed in terms of the following objective:

$$D(\pi, \pi_E)_{APC} = \mathbb{E}_{(a,s) \in \mathcal{B}_E} [\log \pi(a|s) + \mathbb{E}_{\delta s \sim \mathcal{N}(0, \sigma_s^2), a' \sim \pi_E(\cdot|s + \delta s)} \log \pi(a'|s + \delta s)] \tag{10}$$

We call this approach *Augmented Policy Cloning* (APC) as it queries the expert policy to augment the training data. This approach is different from a naive data-augmentation technique, where a new state would be generated, but associated with the original action (and not a new one). It therefore allows to build policies which are feedback-responsive with respect to the expert. We illustrate it in Figure 1 and we formulate APC algorithm for BC in Algorithm 1.

---

**Algorithm 1** Augmented Policy Cloning (APC)

---

$\pi_\theta$ - parametric student policy, $\theta_0$ - initial parameters, $\pi_E$ - expert policy, $\alpha$ - learning rate
$\sigma_s$ - state perturbation noise, $M$ - # augmented samples $K$ - # gradient updates, $L$ - batch size
Dataset $\mathcal{B}_E = \{(s_i, a_i), i = 1, \ldots, N\}, N > 0$ of expert state-action pairs
**for** k=1,…,K **do**
    Sample a batch of pairs $\{(a_i, s_i)\}_{i=1}^L \sim \mathcal{B}_E$
    For each state $s_i$, sample $M$ perturbations $\delta s_j \sim \mathcal{N}(0, \sigma_s), j = 1, \ldots, M$
    Construct $M$ virtual states $s'_{i,j} = s_i + \delta s_j, i = 1, \ldots, L, j = 1, \ldots, M$
    Resample new actions from expert $a'_{i,j} \sim \pi_E(\cdot|s'_{i,j})$
    For Gaussian experts, the action $a_i = \mu_E(s_i)$ and the new actions are $a'_{i,j} = \mu_E(s'_{i,j})$
    Compute the empirical negative log-likelihood:
$$\mathcal{L} = - \left[ \log \pi_{\theta_k}(a_i|s_i) + \frac{1}{M} \sum_{j=1}^M \log \pi_{\theta_k}(a'_{i,j}|s'_{i,j}) \right]$$
    Update the parameters $\theta_{k+1} = \theta_k - \alpha \nabla_\theta \mathcal{L}$
**end for**

---

# 4 Experimental details

In this section we provide details common to all experiments. We provide additional details for each set of results at the beginning of Section 5 and Section 6.

## 4.1 Domains

To study how our method performs on complex control domains, we consider three complex, high-DoF continuous control tasks: *Humanoid Run*, *Humanoid Walls* and *Insert Peg*. All these domains

are implemented using the MuJoCo physics engine [Todorov et al., 2012] and are available in the `dm_control` repository [Tunyasuvunakool et al., 2020]. These problems are rather challenging, requiring stabilization of a complex body (for humanoid tasks), vision to guide the movement (*Walls* task), and solving a complex control problem with a weak reward signal (*Insert Peg*). These environments are related to the domains that have been proposed for use in offline RL benchmarks [Gulcehre et al., 2020]; however, the experiments we perform in this work require availability of the expert policy, so we do not use offline data, but instead train new experts and perform experiments in the very low data regime. We compare all methods on *Humanoid Run* and *Humanoid Walls* tasks and report report a subset of results on *Insert Peg*, due to complex nature of the experiments.

## 4.2 Baselines

As baselines we consider simple BC as described in eqn. (3) as well as a simple modification of BC, where, similarly to APC, we apply state perturbations to expert trajectories as in eqn. (7) and eqn. (8), but we do not produce a new action from the expert (i.e., we augment the states but keep the same action). We call this approach Naive Augmented Behavioral Cloning (Naive ABC). Essentially, this method trains a student policy to produce the same action in response to small state perturbations. This approach is motivated by analogy to how one might build robustness in a classifier. However, this is naive when applied to continuous control problems where even small changes in input should lead to a change in action. Moreover, for *Humanoid Walls* task, we considered additional random crops augmentations applied to visual input of the student (not the expert) which is similar in spirit to Laskin et al. [2020]. Note that in this case, it would also robustify the student to these vision augmentations as it will not produce a new action even in the APC (since the expert was not trained with data augmentations). When vision augmentations are used together with either APC or naive ABC, we add "with image" to the method name. When only image augmentations are used (without any state-based augmentations), we call it "image only". The purpose of combining vision and state augmentations is to study the interplay between APC and more traditional data augmentation methods. We only report results with vision augmentations for *DAgger* and *kickstarting* in the main paper and we provide additional offline policy cloning results in Appendix 8.

## 5 Core results: offline policy cloning

**Training and evaluation protocols.** We train expert policies till convergence using MPO algorithm Abdolmaleki et al. [2018] for *Humanoid* tasks and VMPO algorithm Song et al. [2019] for *Insert Peg* task, as we found MPO was unable to learn on this task. The policies are represented by the Gaussian distribution $\pi_E(\cdot|s) = \mathcal{N}(\mu_E(s), \sigma(s))$. We create datasets as in eqn. (3) using pre-trained experts with a different number of expert trajectories. To asses the sensitivity of different methods to the expert noise, when constructing a dataset, the expert action is drawn according to Gaussian distribution with a fixed variance, i.e. $a \sim \mathcal{N}(\mu_E(s), \sigma_E)$, where $\sigma_E$ is the fixed amount of expert noise. In the subsequent BC experiments, we use $\sigma_E = 0.2$. Moreover, in order to analyze the noise robustness of the student policy is trained via BC, $\pi(\cdot|s) = \mathcal{N}(\mu(s), \sigma(s))$, we evaluate it by executing the action drawn from a Gaussian with a fixed variance, i.e. $a \sim \mathcal{N}(\mu(s), \sigma)$, where $\sigma$ is the fixed amount of student noise. In all the experiments we use $\sigma = 0.2$. We apply early stopping and select hyperparameters based on the evaluation performance on a validation set. We always report performance based on 150 random environment instantiations. For more details, see Appendix 4.

### 5.1 Applying Augmented Policy Cloning

We evaluate the performance of APC when fitting the fixed dataset of expert trajectories. For the APC method, we rely on Algorithm 1. We use baselines described in Section 4.2. In Figure 2, we show the performance of different methods on different tasks as a function of number of trajectories available in the dataset. We see that APC needs significantly fewer expert trajectories to achieve a high level of performance. Moreover, we see that Naive ABC performs very similarly to BC. The results suggest that when cloning an expert using a small fixed dataset APC can provide significant advantages.

In Appendix 6, we report additional results for a scenario, where instead of full long trajectories for each task, we consider only short trajectories (i.e., the early portion of episodes). The motivation for this experiment is to see whether we can further improve the data efficiency of the methods. Incidentally, this supplemental comparison shows that for *Insert Peg* all methods performed better

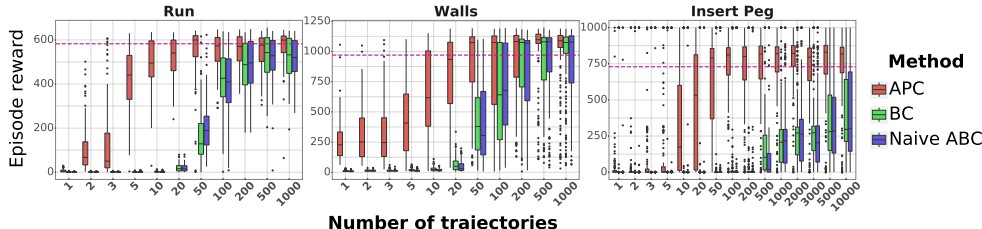

Figure 2: **Offline expert cloning results**. The X-axis represents the number of trajectories, the Y-axis corresponds to the episodic reward across 150 independent evaluations. The red color corresponds to APC, the blue to Naive ABC and the green to BC. The pink line depicts the average performance of the teacher policy. Each subplot represents a different task.

with short trajectories, because initial snippets of episodes actually include the full solution to the task (i.e., the expert policy rapidly inserts the peg and the episode doesn't immediately terminate). We report only long trajectories in Figure 2, since APC performs relatively well in both cases and full length trajectories correspond to the most straightforward setting.

## 5.2 Expert compression

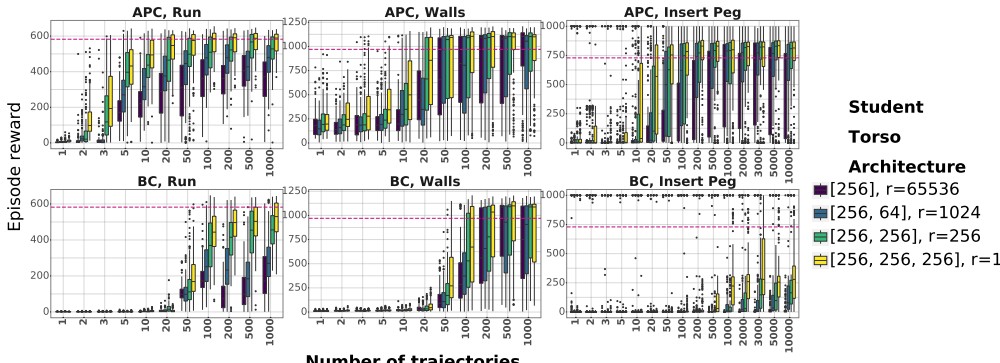

Figure 3: *Teacher compression* **results**. We plot performances of different methods (rows) on different tasks (columns) as a function of number of trajectories available in the expert dataset. The legend corresponds to different student architectures used as well as the relative difference in parameter count in a baseline torso architecture compared to the student torso architecture. We see that the performance degrades much more drastically for BC compared to APC. The pink dashed line corresponds to the performance of the expert policy. We report more complete results in Appendix 8.1

To study APC in a practically motivated setting we consider *expert compression* as discussed in Section 2.3, where a student policy has fewer parameters than the expert. This setting occurs, for instance, when the system is subject to computational contstraints (time, memory, etc.), as in [Parisotto and Salakhutdinov, 2021]. To study APC's data efficiency in this setting, we consider different sizes of the student network torso, where $[256, 256, 256]$ corresponds to the original network size (see Appendix 4.1 for more details). We define the relative difference in parameter count in baseline torso with respect to the student architecture torso $r$ as the product of a baseline torso dimensions, i.e. $256^3$ divided by the student torso dimensions (i.e., for a student torso $[256, 256]$, it is equal to $256^2$).

The results are given in Figure 3. We observe that the performance of all methods degrades when the student network is smaller than the original one, but the degradation is much less severe for APC, while maintaining high level of data efficiency. See Appendix 8.1 for more complete results (with naive ABC) as well as additional ablation over student network torso sizes.

## 5.3 Learning from privileged experts.

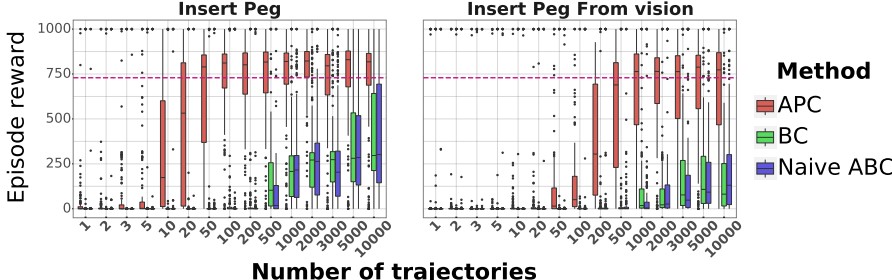

Figure 4: **Learning from privileged experts results**. On the left plot we present the performance of different methods on *Insert Peg* task where additional privileged information (target position) is available. On the right, we present the results where instead of privileged information (target position), we use visual input. We observe that learning from vision is less data efficient and more complicated, but APC manages to obtain comparable performance to the scenario with privileged information, whereas BC and Naive ABC fail to learn. The dashed pink line corresponds to the performance of the expert policy.

Next, we consider a scenario where the expert has access to *privileged information* that is not available to the student, as discussed in Section 2.4. To study the impact of APC in this scenario, we train the expert on *Insert Peg* task where the full state contains common information (proprioception, sword position and orientation) and privileged information of the target position. The student is given access to the common observations as well as a third person (camera) view of the scene providing information about the target position. The latter setup is similar in spirit to [Laskin et al., 2020] For more details, see Appendix 4.2. From Figure 4, we observe that APC achieves similar performance in both settings, provided a sufficient amount of trajectories are available, whereas BC and Naive ABC fail to transfer the expert's behavior to the student.

## 6 Additional Results: Augmented Policy Cloning as a subroutine

### 6.1 DAGGER with data augmentation

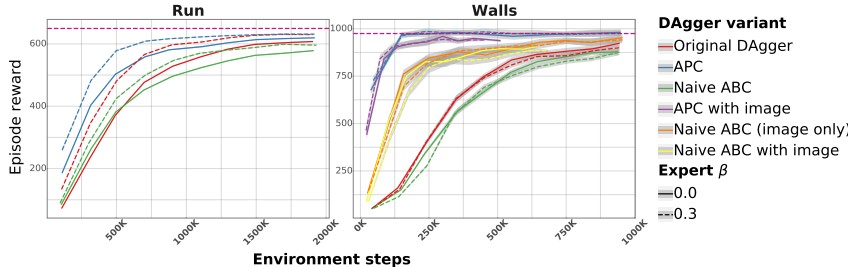

Figure 5: **DAGGER results**. On the X-axis we report the number of environment steps. On the Y-axis we report averaged across 3 seeds episodic reward achieved by the student. The legend corresponds to the DAGGER variant with additional data augmentation is applied. We report confidence intervals in the shaded areas. For Run task, the confidence intervals are very small and are not visible. In solid line we report the performance without using expert policy during the acting. In dashed line, we report the performance of the policy which mixes 30% with the expert. The dashed pink line corresponds to the performance of the expert policy trained via MPO.

As described in Section 2.5, DAGGER [Ross et al., 2011] is a more sophisticated approach where data is collected from the real environment by executing a policy from eqn. (5), which is a mixture between a student and an expert. In this section we study how data augmentation approaches affect the data efficiency of the DAGGER algorithm. For each task, we train expert policies to convergence using the MPO algorithm Abdolmaleki et al. [2018]. We consider similar baselines for both tasks

as in the previous section. For an expert policy that has been pre-trained via MPO [Abdolmaleki et al., 2018], we perform online rollouts for two values of the expert-student mixing coefficient, $\beta = 0$ and $\beta = 0.3$ (see eqn. 5). Since both student and expert are Gaussian distributions, instead of using a $\log \pi$ in eqn. (4), we could use a state-conditional cross entropy from an expert to a student, $\mathcal{H}[\pi_E(\cdot|s)||\pi(\cdot|s)]$. Empirically, we found that it worked better than using $\log \pi$ (see Appendix 8.6). We run experiments in a data-restricted setup. For more details, see Appendix 5.1.

Results are shown in Figure 5. We report the performance of both the original DAGGER as well as DAGGER with a data augmentation. We see that APC and its vision variant outperform BC and Naive ABC similarly to the behavior cloning experiments. While we observe that image augmentation can help, we see that the primary advantage comes from the state-based augmentation for APC. For the Run task, we observe that all DAGGER methods achieve slightly lower performance than an expert policy. We speculate that this is due to insufficient coverage of the state space during training.

## 6.2   Kickstarting with data augmentation.

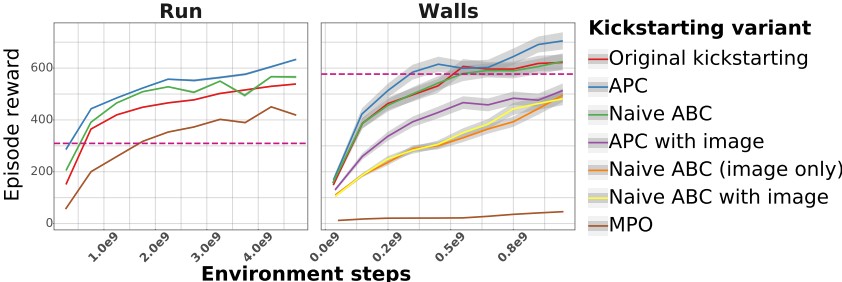

Figure 6: **Kickstarting results**. On the X-axis we show the number of environment steps and on the Y-axis we report averaged across 3 seeds episodic reward achieved by the student. The legend corresponds to the kickstarting variant where additional data augmentation is applied. We report confidence intervals in the shaded areas. For Run task, these intervals are small and are not visible. Dashed pink line shows the expert performance.

A similar in spirit approach is kickstarting Schmitt et al. [2018], where we solve an RL task as well as cloning the expert policy. Similarly to previous section, we apply APC in kickstarting on the cross entropy term in eqn. (6). For each task, we train expert policies to convergence using the MPO algorithm Abdolmaleki et al. [2018]. Since in the kickstarting we are interested in outperforming a sub-optimal expert, for each task, we select experts such that they achieve around 50 % of the optimal performance. On top of kicktarting, we report the performance of MPO Abdolmaleki et al. [2018] learning from scratch on the task of interest. We run experiment in a distributed, high data regime. All the details are given in Appendix 5.2.

The results are given in Figure 6. We report the performance of both the original kickstarting as well as kickstarting with additional data augmentation. We observe that APC performs better than Naive ABC on *Humanoid Run* task and similarly on *Humanoid Walls* task. Both approaches perform better than BC and learning from scratch. We hypothesise that the reason of not seeing a consistent advantage could be due several factors. Firstly, as we are in a high-data and distributed regime, since there is no limit on relative acting / learning ratio, and acting policies are not restricted to collect trajectories, it is unclear whether data-augmentation should help. We tried to explore the rate-limiting regime, but we experienced instability of kickstarting experiments. Secondly, we use reward signal which makes the impact of expert cloning less important. Thirdly, on top of learning the policy, we also need to learn an state-action value $Q(s, a)$ function. Unfortunately, we cannot use APC-style data augmentation for learning $Q$ function, therefore it might be the bottleneck. Finally, unlike in kickstarting Schmitt et al. [2018], we do not use an annealing schedule of $\lambda$ to make the experiments simpler, but we still observe that a fixed coefficient helps to kickstart an experiment and outperform an expert policy. On top of that, we see that image-based augmentation have less of impact in this setting and generally leads to poor performance.

# 7 Discussion

Many expert-driven learning approaches actually have access to an expert that can be queried; however, this opportunity is rarely exploited fully. In this work we demonstrated a general scheme for more efficient transfer of expert behavior by augmenting expert trajectory data with virtual, perturbed states as well as the expert actions in these virtual states. This data augmentation technique is widely applicable and we demonstrated that it improves data efficiency when used in place of behavioral cloning in various settings including offline cloning, expert compression, transfer from privileged experts, or when behavioral cloning is used as a subroutine within online algorithms such as DAGGER or kickstarting. One could argue that other way to clone the expert would be to query it on densely covered state space. Such an approach, however, would require exponentially more number of queries as the dimensionality of the state space grows. Our approach, however, would scale similarly to baseline methods (such as DAGGER or Behavioral Cloning) as it would require only $M$ additional queries for each state during learning (see Algorithm 1). In case when the queries to the expert are expensive, our approach may become problematic.

Critically, data efficiency is generally very important in realistic applications, where new data acquisition cost could be high. In particular, settings involving deployment of policies in the real world, such as robotics applications, may benefit from an ability to efficiently transfer expert policy behavior from one neural network to another (for compression or execution speed reasons). While overall, we consider the present work to be fairly basic research with limited ethical impact, insofar as our approach decreases the amount of data which needs to be collected through processes which could potentially be unsafe or costly, there is a potential positive social value.

Our approach is neither intended for nor suitable for all control settings. Fundamentally, our approach relies upon the ability to query expert policy for the perturbed states. This arises frequently enough to be worth our investigation, but is a limiting assumption. Our approach was also developed with continuous control problems, essentially with continuous observation spaces as well as continuous action spaces in mind. Related approaches may be worth pursuing in discrete control problems, but that has not been a focus of the present work.

In future work, it may be interesting to develop an extension of our approach that is compatible with offline RL. While offline RL operates with different assumptions, i.e., we generally do not have access to the policy which produced the data, we may wish to consider adjacent scenarios where the queryable expert can be combined with offlline learning RL or where augmentations facilitate learning the Q-function.

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
