# Data augmentation for efficient learning from parametric experts: Supplementary material

## 1 Environment details

In this work we consider three environments from DeepMind Control repository [Tunyasuvunakool et al., 2020]: *Humanoid Run*, *Humanoid Walls* and *Insert Peg*. *Humanoid Run* task requires an agent controlling humanoid body to run at a specific speed and gets reward which is proportional to the inverse distance between its current speed and the target speed. The observations are based on proprioceptive information. In *Humanoid Walls*, an agent controls a humanoid body to run along a corridor and avoid walls, at highest possible speed. The observations are based on proprioception and on egocentric vision. It receives reward which is proportional to the forward speed (through the corridor), thus incentivising it to run as fast as possible. It receives proprioceptive observations as well as the image of size 64x64 from the ego-centric camera. In our experiments, we use *Simple Humanoid* body rather than *CMU Humanoid* as in the original task in order to simplify the experiments. Action dimension is equal to 21. Finally, in *Insert Peg*, an agent controls an arm which needs to put a sword into a narrow hole. The observations are based on proprioception, sword position and orientation, hole position. In case of vision-based input, we add second person camera observations with image size of 64x64. For more details, check Tunyasuvunakool et al. [2020]. Note these environments are related to the domains that have been proposed for use in offline RL benchmarks [Gulcehre et al., 2020]; however, the experiments we perform in this work require availability of the expert policy, so we do not use offline data, but instead train new experts and perform experiments in the very low data regime. We choose these exact environments due to their challenging nature in order to demonstrate the impact of APC on data efficiency, but we did try initially simpler environments from the DeepMind Control repository [Tunyasuvunakool et al., 2020] which included *Humanoid Walk* and *Walker Walk/Run* tasks. We decided not to conduct exhaustive experiments on these environments.

## 2 Methods and baselines

The main method we consider is APC described in Section 3 from main paper for offline experts cloning experiments. For all the methods in offline expert cloning experiments, as an action in the objective from eqn. (3) from main paper, we use an expert mean $\mu_E(s)$. We also extend this method to scenarios from Section (6) from main paper, i.e. on DAGGER and kickstarting. For both scenarios, applying APC is straightforward. In case of DAGGER, the APC approach would correspond to resampling additional states via eqn. (7) from main paper and via eqn. (8) from main paper for each state $s_t$ encountered in the objective from the eqn. (4) from main paper. Then, for each such new state $s' = s_t + \delta s$, we would add a term to optimize in the objective, corresponds to the cross entropy form the expert to the student, i.e. $\mathcal{H}[\pi_E(\cdot|s')||\pi(\cdot|s')]$. We can also consider resampling a new action, but we empirically found that cross-entropy worked better, see Appendix (8.6). For kickstarting, applying APC would also correspond to sampling new virtual states $s' = s + \delta s$ for each state in the second term of the eqn. (6) from main paper. Then similarly, we would add an additional cross entropy term to the objective, i.e., $\mathcal{H}[\pi_E(\cdot|s')||\pi(\cdot|s')]$.

As baselines against APC, we consider BC algorithm described in eqn. (3) from main paper, which in DAGGER and kickstarting simply corresponds to the unmodified versions of this method. On top of BC, we consider a simple modification of BC, where we apply, similar to APC, state perturbations to expert trajectories as in eqn. (7) from main paper and eqn. (8) from main paper, but we do

36th Conference on Neural Information Processing Systems (NeurIPS 2022).

not produce a new action from the expert and use the original one. We call this approach Naive Augmented Behavioral Cloning (Naive ABC). Essentially, this method trains a student policy to be robust with respect to small state perturbations. The application of Naive ABC in case of DAGGER and kickstarting is similar to APC, with the exception that we now consider the cross entropy term $\mathcal{H}[\pi_E(\cdot|s)||\pi(\cdot|s')]$, where the student is taken on the new augmented states and the expert on the original, unmodified ones.

Moreover, for *Humanoid Walls* task, we considered additional vision-based augmentations, random crops, similar in spirit to Laskin et al. [2020]. Note that in this case, it would also robustify the student to these vision augmentations as it will not produce a new action even in the APC (since the expert was not trained with data augmentations). When vision augmentations are used together with APC or naive ABC, we add "with image" to the method name. When only image augmentations are used (without any state-based augmentations), we call it "image only". The purpose of combining vision and state augmentations is to study the interplay between APC and more traditional data augmentation methods. We use random crops producing images of size 48x48 instead of the original 64x64 images.

## 3 Agent architecture

For all the experiments, we use the same agent architecture. The agent has two separate networks: actor (policy) and critic (Q-function). Both networks are split into 3 components: encoder, torso and head. Encoders for actor and critic are separate but have the same architecture. For state-only (no vision) observations, encoder corresponds to a simple concatenations of all the observations. For the visual input, it divides each pixel by $255$ and then applies a 3-layer ResNet of sizes $(16, 32, 32)$ with $ELU$ activations followed by a linear layer of size $256$ and $ELU$ activation. The resulting output is then concatenated together with state-based input. For actor network, torso corresponds to a 3 dimensional MLP, each hidden layer of size $256$ with activation $ELU$ applied at the end of each hidden layer. The output of actor torso is then passed to the actor head network, which applies a linear layer (without activation) with output size equal to $N_a * 2$, where $N_a$ is the action dimension. It produces the actor mean $\mu$ and log-variance: $\log\tilde{\sigma}$. Then, the variance of the actor is calculated as

$$\sigma = softplus(\log\tilde{\sigma}) + \sigma_{min},$$

where $\sigma_{min} = 0.0001$. That would encode the Gaussian policy $\pi(\cdot|s) = \mathcal{N}(\mu(s)|\sigma(s))$. This parameterization ensures that the variance is never 0. The critic torso network is 1 dimensional MLP of size $256$ with $ELU$ activation on top of it. Both critic encoder and critic torso are applied to the state input and not the action. The output of torso and the action are passed to the head, which firstly applies a $tanh$ activation to the action to scale it in $[-1, 1]$ interval, then concatenates both scaled action and torso output. This concatenated output is then passed through a 3-dimensional MLP with sizes $[256, 256, 1]$ with $ELU$ activations applied to all layers except the last one. This produces the Q-function representation, $Q(s, a)$. The critic network is not used for the offline expert cloning and DAGGER experiments.

To train experts for *Humanoid Run* and *Humanoid Walls* tasks, we use MPO [Abdolmaleki et al., 2018] algorithm with default hyperparamertes. For *Insert Peg* experiments, we use VMPO [Song et al., 2019] algorithm since we found that MPO [Abdolmaleki et al., 2018] failed to train. We use the same architecture as described above and use default hyperparameters from VMPO original paper.

## 4 Offline policy cloning experiment details

For each task, we train expert policies till convergence. We use MPO algorithm Abdolmaleki et al. [2018] for *Humanoid* tasks and VMPO algorithm Song et al. [2019] for *Insert Peg* task, as we found MPO was unable to learn on this task.

The policies are represented by the Gaussian distribution $\pi_E(\cdot|s) = \mathcal{N}(\mu_E(s), \sigma(s))$. We create datasets as in eqn. (3) from main paper using pre-trained experts with a different number of expert trajectories. To asses the sensitivity of different methods to the expert noise, when constructing a dataset, the expert action is drawn according to Gaussian distribution with a fixed variance, i.e.

$$a \sim \mathcal{N}(\mu_E(s), \sigma_E), \tag{1}$$

where $\sigma_E$ is the fixed amount of expert noise. We consider 4 different levels of $\sigma_E$: **Deterministic**, meaning that we unroll the expert trajectories using only the mean $\mu_E$, **Low**: $\sigma_E = 0.2$, **Medium**: $\sigma_E = 0.5$, **High**: $\sigma_E = 1.0$. We also tried values in-between, but did not found a qualitative difference. We also tried values above $\sigma_E = 1.0$, but the performance for these ones was almost zero. In all the experiments, we use $\sigma_E = 0.2$. We provide additional ablation over different levels of expert noise $\sigma_E$ in Appendix 8.3.

We unroll the expert trajectories by chunks containing 10 time steps each and put it in a dataset. We use Reverb (from ACME [Hoffman et al., 2020]) backend for this. A full trajectory for a *Humanoid Run* task corresponds to 1000 time steps which corresponds to 25 seconds of control time with a control discretization of 0.025 seconds. A full trajectory for a *Insert Peg* task corresponds to 1000 time steps which corresponds to 10 seconds of control time with a control discretization of 0.1 seconds. A full trajectory for the *Humanoid Walls* task corresponds to 2000-2500 time steps. This variation is due to potential early stopping of the task execution (in case if the agent falls down). The discretization for the control is 0.03 seconds and maximum episode length is 45 seconds.

For each task, we construct datasets containing 1, 2, 3, 5, 10, 20, 50, 100, 200, 500, 1000 trajectories. For *Insert Peg* task, we also create datasets containing 2000, 3000, 5000 and 10000 trajectories, as we found this task requiring more data to be able to be learned.

When evaluating the method, in order to analyze the noise robustness of the student policy is trained via BC, $\pi(\cdot|s) = \mathcal{N}(\mu(s), \sigma(s))$, we evaluate it by executing the action drawn from a Gaussian with a fixed variance, i.e.

$$a \sim \mathcal{N}(\mu(s), \sigma), \tag{2}$$

where $\sigma$ is the fixed amount of student noise. We tried similar value for $\sigma$ as in case of expert noise $\sigma_E$. In all the experiments below we use $\sigma = 0.2$. We provide additional ablation over these values in Appendix 8.3.

To train offline expert cloning methods we rely on Algorithm 1 as the main algorithm for all the methods, where we remove additional action or/and state augmentations for Naive ABC and BC. For all the experiments we use learning rate $\alpha = 0.0001$, number of augmented samples $M = 10$, batch size $L = 64$. We did try different values for these parameters and found it made no difference on the performance and final experiments outcome, so we fixed one set of parameters for the simplicity of the experimentation.

For every method and task, we tune method-specific hyperparameters. It corresponds to tuning state noise perturbation variance $\sigma_s^2$ from eqn. (7) from main paper for APC and Naive ABC (we do not need to tune it for BC as we do not use any data augmentation in case of BC). The considered range for this parameter is $[0.0001, 0.001, 0.01, 0.05, 0.1, 1.0, 10.0]$. For APC, we found that $\sigma_s = 0.1$ worked best for *Humanoid Run* and *Insert Peg*, and $\sigma_s = 1.0$ worked best for *Humanod Walls* task. For Naive ABC, these values are: $\sigma_s = 0.001$ for *Humanoid Run*, $\sigma_s = 0.01$ for *Insert Peg* and $\sigma_s = 0.0001$ for *Humanoid Walls* tasks. We select these parmameters based on the validation set performance in the procedure described in the next paragraph. We use the same values of $\sigma_s$ for vision-based augmentation variants as the original method to which these vision-based augmentations are applied. For example, if the method is APC with image, it means that we use $\sigma_s$ for APC in this experiment.

We train all the offline expert cloning methods till convergence (maximum 20M iterations) for each experiment / task / method variant. Each iteration corresponds to applying gradients to the batch of 64 trajectories, each containing 10 time steps. We evaluate each model using 150 random environment instantiations. We noticed that when we train models offline, at convergence there are small variations in performance among subsequent models. Therefore, we use early stopping to select for the best model for each experiment. In order to do that, we use a validation set of separate 50 random environment instantiations and we select the best model based on the average performance among these 50 instantiations. We use the same early stopping procedure to select for the best hyperparameter.

## 4.1 Expert compression: details

We consider *expert compression* setting as discussed in Section 2.3 from main paper, where a student policy has smaller parameters than the expert. This setting often occurs in situations where there are computation constraints (memory, etc.) on the system which would be used on the student as in

[Parisotto and Salakhutdinov, 2021]. To study the APC data efficiency in this setting, we consider different sizes of the student network torso, where $[256, 256, 256]$ corresponds to the original network size. In particular, we consider following sizes: $[256], [256, 64], [256, 256]$. We consider additional network sizes and provide additional ablations in Appendix 8.1.

## 4.2 Learning from privileged experts: details

We consider a scenario where the expert has access to the *privileged information* whereas student does not, as discussed in Section 2.4 from main paper. Typically, in such a scenario, it is easier to train the expert than the student, but training a student with a restricted observations is more preferable in an application.

To study the impact of APC in this scenario, we train the expert on *Insert Peg* task where the full state contains common information (proprioception, sword position and orientation) and privileged information of the target position. The student is then trained on the common observations and on vision-based input through the second person camera which replaces privileged information. The latter setup is similar in spirit to [Laskin et al., 2020]. The student network with additional vision observations therefore has an additional visual input encoder as described in Appendix 3. It encodes non-vision observations with simple concatenation and concatenates the result with the vision embedding. It is then passed through the same torso and head networks as original expert. In this regime, the student does not know about the target position and needs to infer it from vision-based observations.

## 5 APC as subroutine: experimental details

### 5.1 DAGGER experiment details

For each task, we train expert policies to convergence using the MPO algorithm Abdolmaleki et al. [2018]. Each expert is represented by a Gaussian policy, see Appendix 3. Throughout the experiment, we use the replay buffer of size $1e6$ where each element corresponds to 10-step trajectory, implemented using Reverb (from ACME [Hoffman et al., 2020]). We use the actor-learning architecture, with 1 actor and 1 learner, where the actor focuses on unrolling current policy and on collecting the data, whereas the learner samples the trajectories from the replay buffer and applies gradient updates on the parameters. When doing so, we control a relative rate of acting / learning via rate limiters as described in Hoffman et al. [2020] such that for each time step in the trajectory, we apply in average 10 gradient updates. This allows us to be very data efficient and get the full power from the data augmentation technique. In order to achieve it, we set the samples per request (SPI) parameter of the rate limiter to be $T * B * 10$, where $T = 10$ is the trajectory length (sample from a replay buffer), $B$ is the batch size (256 for Run and 32 for Walls). When sampling from the replay buffer, we use uniform sampling strategy. When the replay buffer is full, the old data is removed using FIFO-strategy.

For each method and each domain, we run the experiment with 3 random seeds. Normally, in DAGGER, the parameter $\beta$ of mixing the experience between the student and an expert, should decrease to 0 throughout the learning. For simplicity of experimentation, we used fixed values. We report the results using $\beta = 0$ and $\beta = 0.3$, but we also experimented with values $\beta = 0.1, 0.2, 0.4, 0.5$. We found that our chosen values provided most of the qualitative information. The values of state perturbation noise for APC are: $\sigma_s = 0.1$ for Run and $\sigma_s = 1.0$ for Walls task. For Naive ABC, these values are: $\sigma_s = 0.01$ for Run and $\sigma_s = 0.001$ for Walls tasks. The values which we tried are: $[0.00001, 0.0001, 0.001, 0.01, 0.1, 1.0, 10.0]$.

When collecting the data, we use a mixture of student and expert, which are represented as stochastic policies via Gaussian distributions. For evaluation, we used their deterministic versions, by unrolling only the mean actions.

To train policies via DAGGER, we used analytical cross-entropy between expert and student instead of log probability of student on expert mean actions, as we found that it worked better in practice. We provide qualitative comparison in Appendix 8.6.

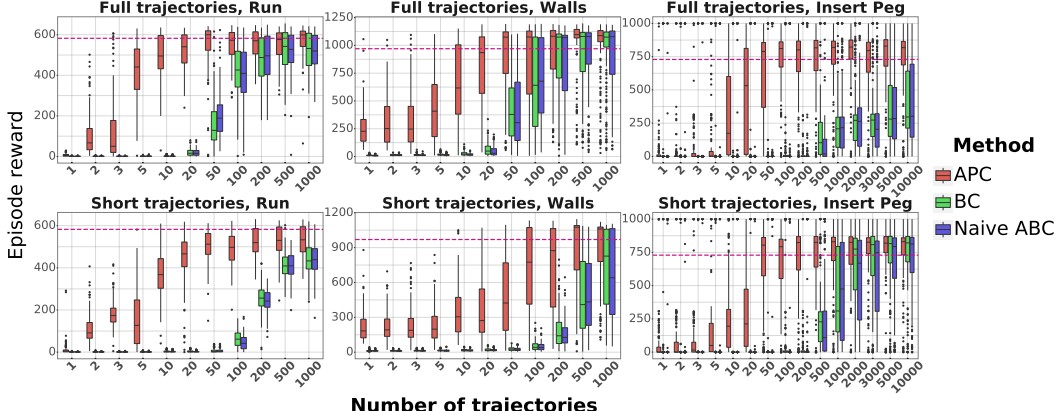

Figure 1: **Offline expert cloning** results with different number of trajectories for APC, BC and Naive ABC on *Humanoid Run*, *Humanoid Walls* and *Insert Peg* tasks represented by columns. First row corresponds to the full trajectory case, as reported in Figure 2 from main paper. Second row corresponds to the case of short trajectories, where the dataset contain one full and a given number of short trajectories. The X-axis represents the number of trajectories, the Y-axis corresponds to the episodic reward across 150 independent evaluations.

## 5.2 Kickstarting experiment details

For each task, we train expert policies to convergence using the MPO algorithm Abdolmaleki et al. [2018]. Since in the kickstarting we are interested in outperforming sub-optimal expert, we select experts which achieve around 50 % of optimal performance on each task. Each expert is represented by a Gaussian policy, see Appendix 3. We run experiments using a distributed setup with 64 actors and 1 learner, which queries the batches of trajectories (each containing 10 time steps) from a replay buffer of size $1e6$. We use Reverb (from ACME [Hoffman et al., 2020] as a backend. Batch size is 256 for Run and 32 for Walls. We run the sweep over $\lambda$ parameter from eqn. (6) from the main paper. As opposed to DAGGER, we do not use the rate-limiter to control the relative ratio between acting and learning as we found that kickstarting in such a regime was unstable. We found that $\lambda = 0.0001$ worked best for Run, whereas $\lambda = 0.01$ worked best for Walls. The values we tried are: $[0.0001, 0.001, 0.01, 0.1, 1.0, 10.0]$. We found that for higher values of $\lambda$, the learning was faster but the resulting policy did not outperform the expert. On top of running BC methods, we also report the performance of MPO Abdolmaleki et al. [2018] learning from scratch on the task of interest. The values of state perturbation noise for APC are: $\sigma_s = 0.01$ for Run and $\sigma_s = 0.01$ for Walls task. For Naive ABC, these values are: $\sigma_s = 0.00001$ for Run and $\sigma_s = 0.0001$ for Walls tasks. The values which we tried are: $[0.00001, 0.0001, 0.001, 0.01, 0.1, 1.0, 10.0]$. On top of that, when using the MPO algorithm during kicktasrting, we modify MPO-specific parameters $\epsilon_\mu$ and $\epsilon_\Sigma$ to $\epsilon_\mu = 0.05$ and $\epsilon_\Sigma = 0.001$ as we found that using higher values for M-step constraints led to better kickstarting performance. When we apply image-augmentations for kickstarting, we only apply it on the student policy and not on student $Q$-function. Empirically, we found that adding image augmentations for $Q$ function inputs led to worse performance.

## 5.3 Plotting details

When we plot the results in Figure (5) from main paper, Figure (6) from main paper, Figure 9 and Figure 10 we use the following method. For each independent task, method and independent run (seed), we split the data into bins, each containing 10% of the data. Then, in each bin, the performance is averaged as well as the 95% confidence interval is calculated. We then report these values in the figure.

## 6 Short trajectories experiment

In this section we present additional results to the ones presented in Section (5) from main paper. We discussed that we construct the dataset of expert trajectories containing full trjaectories (1000

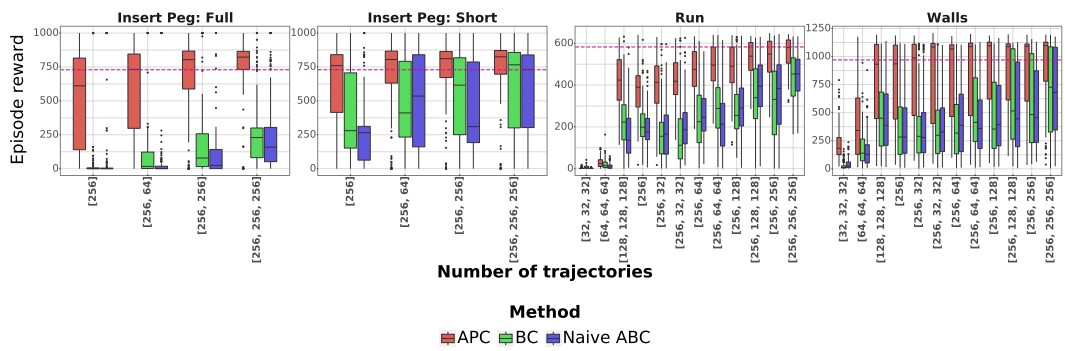

Figure 2: *Teacher compression* results with additional student torso architecture sizes.

timesteps for *Humanoid Run* and *Insert Peg* and around 2000 timesteps for *Humanoid Walls*). It corresponded to a simple unroll of the expert policy on the original environments. In addition to that, we create datasets which contain only one full trajectories and a given number of short trajectories, where each short trajectory contains only 200 timesteps starting from the initial state. The reason for this experiment is to study the ability of different offline expert cloning methods for a more data restricted setup. Note that such a scenario can occur in practice when dealing with realistic robots, where dataset can contain a lot of successful trajectories, but the trajectories can be short, because a robot can fail at some points of time.

We present results in Figure 1, where we duplicate the results for full trajectories and add additional results for short trajectories. We observe that APC performs well in all the cases, whereas BC and Naive ABC performance degrade on *Humanoid Run* and on *Humanoid Walls*. What is interesting, however, is that these methods perform better on *Insert Peg* scenario with short trajectories (but still worse than APC). The reason for is due to the fact that in *Insert Peg*, the rewarding state corresponds to a situation where arm inserted a sword into a hole and does not move (and episode is not finished until 1000 time steps had elapsed). Therefore, long trajectories of expert policies will contain a lot of such states and actions, therefore having a limited diversity. In case of short trajectories, the relative ratio of states preceding this final state is much higher. Interestingly, APC still performs well in both scenarios.

# 7 Total compute used

In total, for all the experiments, we have used machines with GPUs v100 or p100, each machine having 4 CPU cores and 64 Gb of memory. In total, we have ran 2803 experiments.

# 8 Ablations

## 8.1 *Expert compression* additional results

In this section we present additional results for *expert compression* experiment from Section 2.3 from main paper.

Firstly, we present an ablation over different network sizes in Figure 2. We see that generally APC degrades much less than other methods when we decrease the student network size.

Secondly, as an additional to the results in Figure 3 from main paper, we present results for all the tasks and all the methods on Figure 3

## 8.2 Learning from privileged experts: additional results

We present additional results to the experiment presented in Section 2.4 from main paper where we also consider *Insert Peg* tasks where dataset contains only short trajectories as we have seen in Appendix 6 that all methods performed better on this task with short trajectories. The results are given in Figure 4.

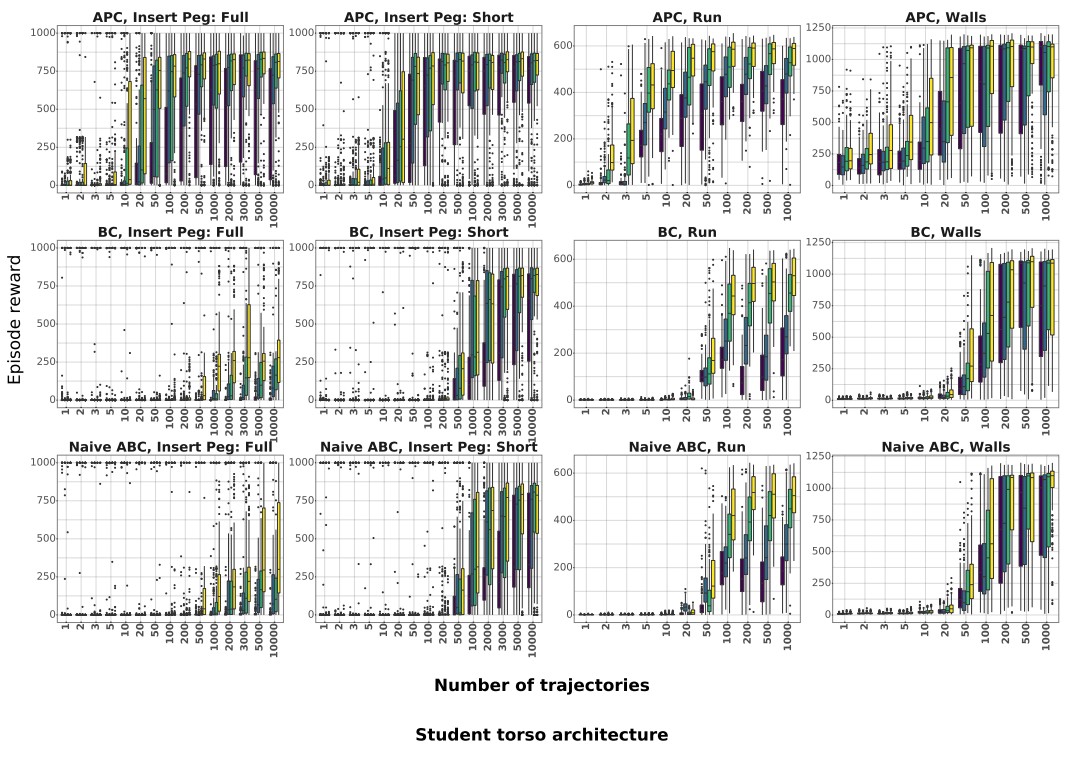

**Student torso architecture**

■[256] ■[256, 64] ■[256, 256] ■[256, 256, 256]

Figure 3: *Teacher compression* with all the methods and tasks. For *Insert Peg*, we consider two setups, with full and short trajectories in the dataset. See Appendix 6

## 8.3 APC expert noise sensitivity

In this section we present additional ablations on the sensitivity of APC, BC and Naive ABC to different values of expert noise $\sigma_E$ from eqn. (1) and student noise $\sigma$ from eqn. (1). We consider four different levels of noise as discussed in Appendix 4. The results are given in Figure 5. For *Humanoid Run* and *Humanoid Walls* tasks, we use 100 trajectories dataset, whereas for *Insert Peg*, we use dataset with 2000 trajectories, as this task is much less data efficient than others. Moreover, for *Insert peg* we consider scenario with full and short trajectories, whereas for *Humanoid* tasks we consider only full trajectories. We see that overall, APC provides more robust policies for different amounts of expert and student noise. For expert noise sensitivity, note that over different columns, the APC performance degrades much less than for BC and Naive ABC. Moreover, for each column, observing the change of the student noise level (from low to high), we see that performance degrades for all the methods, but much less for APC. Therefore, APC seems to provide more action-noise robust policies. We see that BC and Naive ABC perform similarly in terms of robustness. Finally, what is interesting, APC generally observes much less variance in performance when varying the noise levels compared to BC and Naive ABC

## 8.4 APC and ABC state noise ablations

In this section we provide additional ablations for the state-noise perturbation level $\sigma_s$ from the eqn. (7) from the main paper. In Figure 6, we show the results for APC, whereas in Figure 7, we show the results for Naive ABC. We see that there is a sweet spot for the state perturbation noise level.

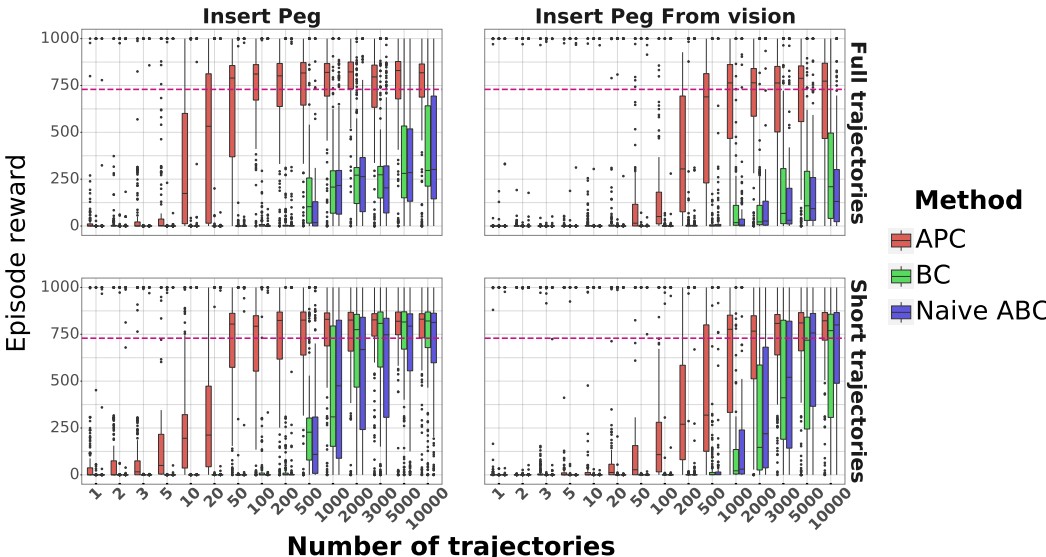

Figure 4: Learning from privileged experts, additional results. On top of the considered results in the main paper, we add additional results with short trajectories.

## 8.5 Additional comparisons for Walls task

In Figure 8, we provide additional results for behavioral cloning experiment on Walls task where we try different variants of APC and Naive ABC with additional image-based augmentation as described in the main paper.

## 8.6 Objective functions comparison for DAGGER

In Figure 9 and in Figure 10, we provide ablations over different objectives for DAGGER with $\beta = 0.0$ and $\beta = 0.3$ correspondingly. We see that overall, training with cross-entropy leads to better results than with log prob on the mean action, especially when $\beta = 0.0$.

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

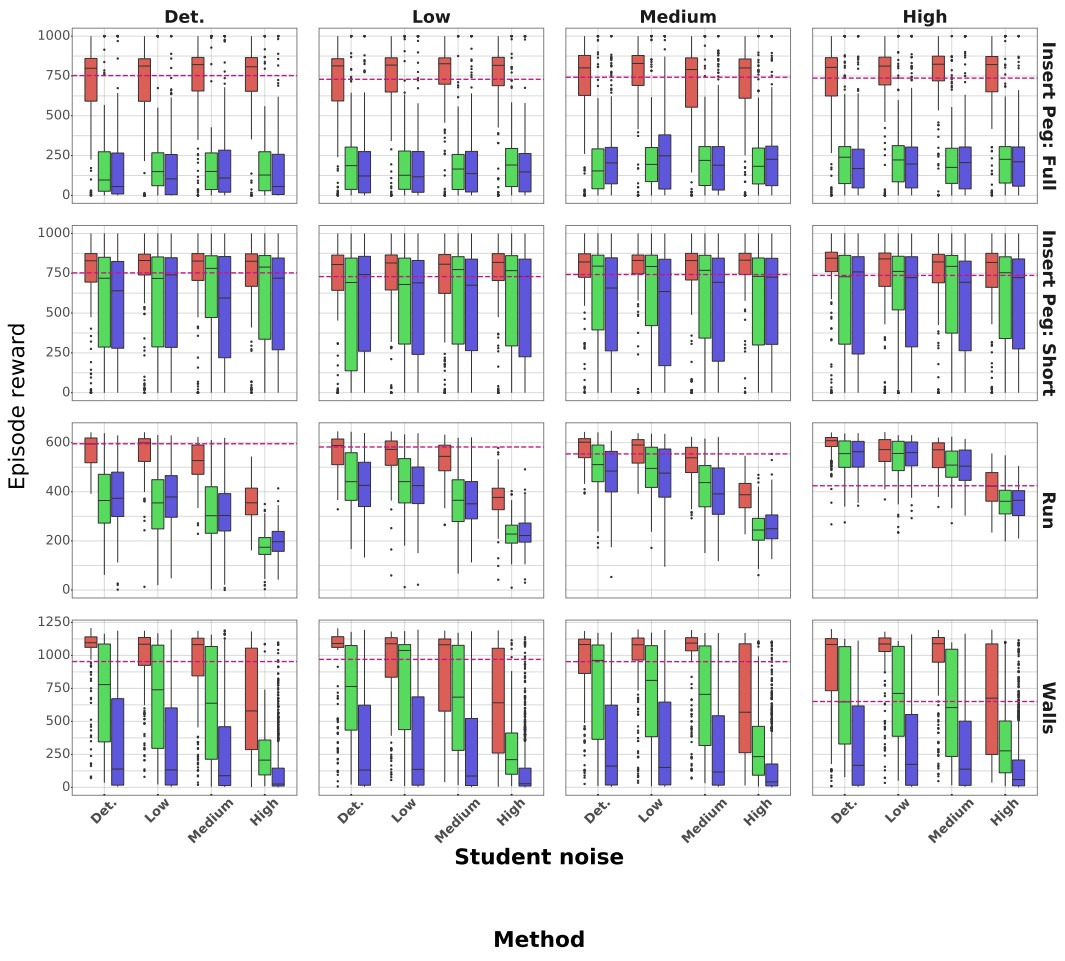

Figure 5: **Noise sensitivity results.** We consider 4 levels of noise for student and expert: **Deterministic**, which uses the Gaussian mean for the action, **Low**, is the noise $\sigma = 0.2$, **Medium** $\sigma = 0.5$ and **High** $\sigma = 1.0$. Each column corresponds to a different level of expert noise. Each row represents different task. For *Insert peg* we consider scenario with full and short trajectories, see Appendix 6. For *Humanoid Run* and *Humanoid Walls* tasks, we use 100 trajectories dataset, whereas for *Insert Peg*, we use dataset with 2000 trajectories, as this task is much less data efficient than others. X-axis corresponds to a different level of student noise. Y-axis corresponds to the episodic reward with 150 independent evaluations. The legend denotes a method and a row corresponds to a task. The pink dashed line indicate average expert performance.

Saran Tunyasuvunakool, Alistair Muldal, Yotam Doron, Siqi Liu, Steven Bohez, Josh Merel, Tom Erez, Timothy Lillicrap, Nicolas Heess, and Yuval Tassa. dm_control: Software and tasks for continuous control. *Software Impacts*, 6:100022, 2020.

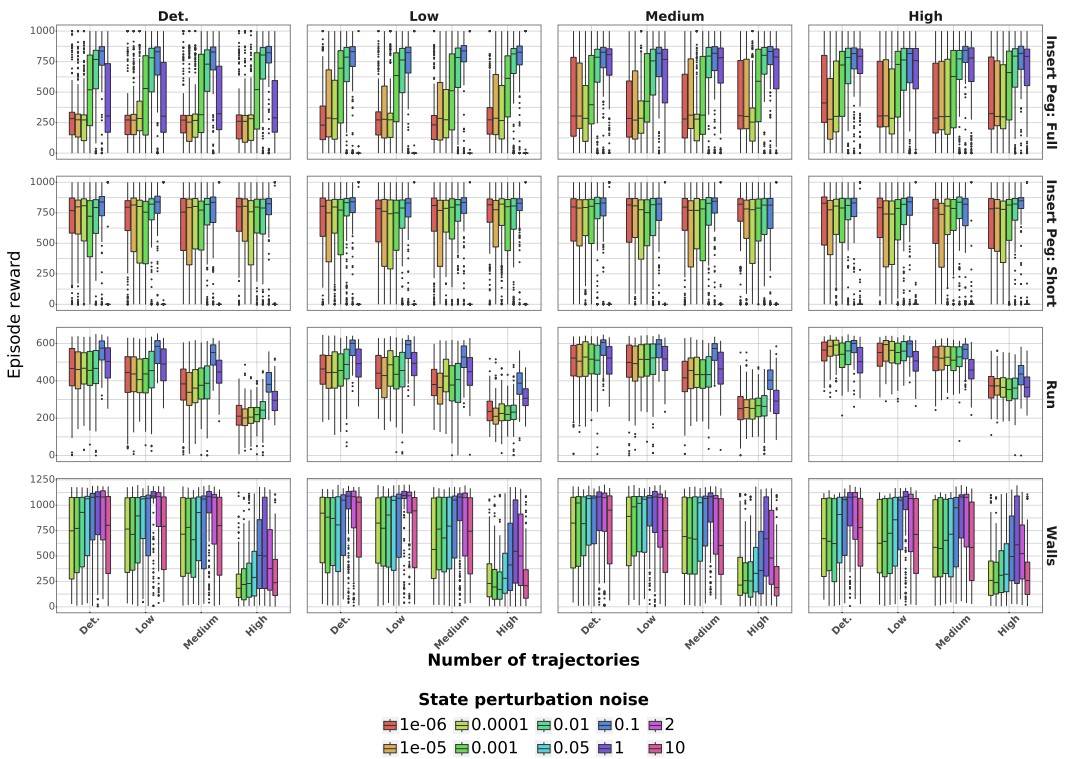

Figure 6: **State perturbation noise sensitivity for APC.** In this plot we represent the APC method trained on 100 full trajectories sampled under different level of expert noise which is represented by different columns. On the X-axis is the different level of a student noise at evaluation time. The legend denotes different levels of a state perturbation noise $\sigma_s$ from the eqn. (7) from the main paper. Y-axis corresponds to the episodic reward with 150 independent evaluations.

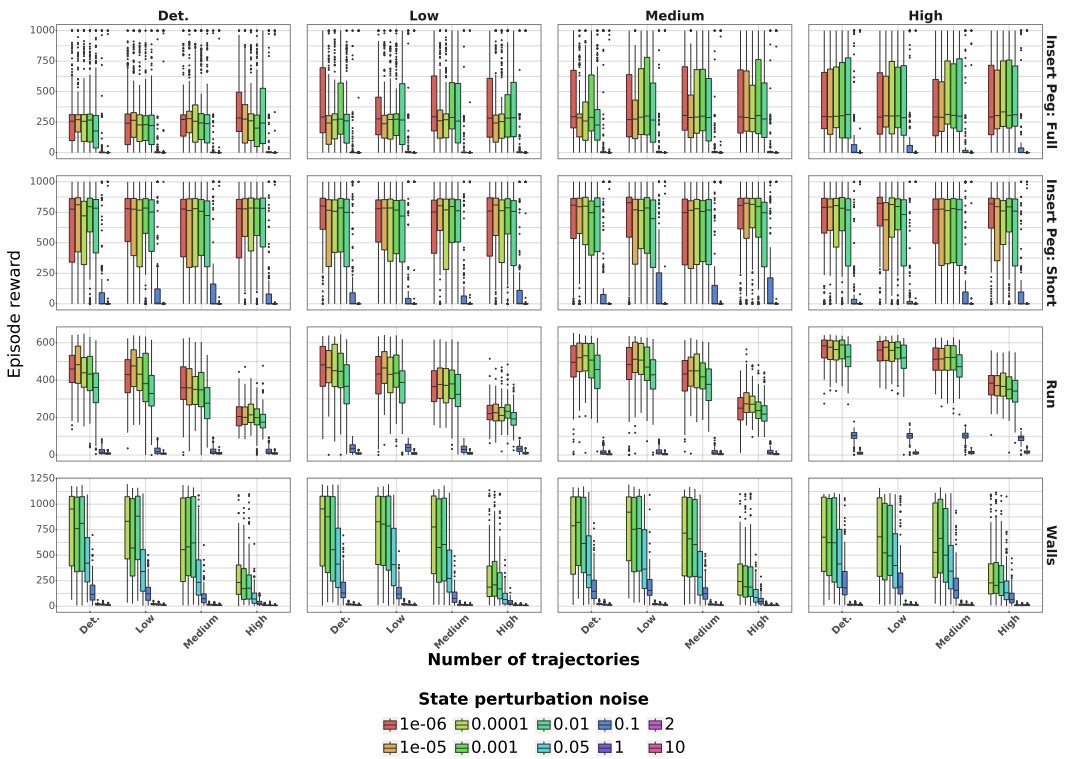

Figure 7: **State perturbation noise sensitivity for Naive ABC**. In this plot we represent the Naive ABC method trained on 100 full trajectories sampled under different level of expert noise which is represented by different columns. On the X-axis is the different level of a student noise at evaluation time. The legend denotes different levels of a state perturbation noise $\sigma_s$ from the eqn. (7) from the main paper. Y-axis corresponds to the episodic reward with 150 independent evaluations.

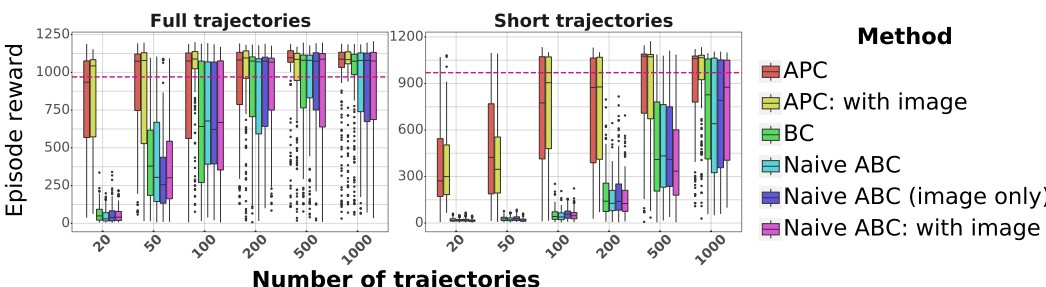

Figure 8: **Additional behavioral cloning results on Walls tasks with additional methods added.** X-axis corresponds to a number of trajectories used in each of the dataset. The Y-axis corresponds to the episodic reward with 150 random evaluations. The pink dashed line indicate average (among the same 150 independent evaluations) expert performance. The legend describes a method which is used. The plot o the left depicts the performance of offline policy cloning with using full trajectories from the expert, whereas the plot on the right represents the experiment with short trajectories. See Appendix 6 for more details.

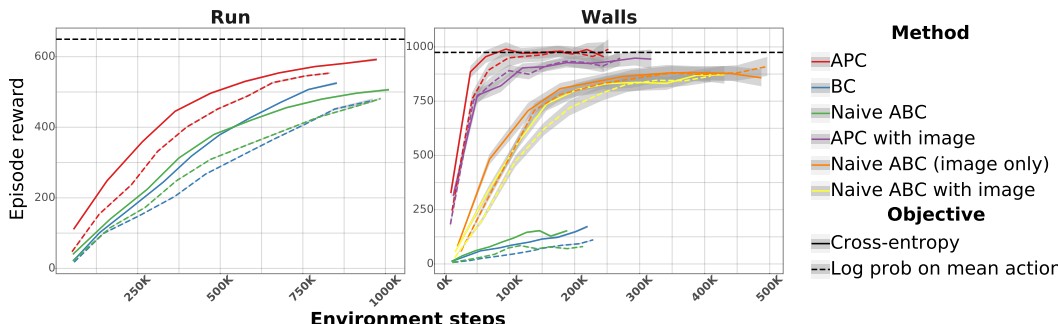

Figure 9: **DAGGER objective sweep with** $\beta = 0.0$. On the X-axis we report the number of environment steps. On the Y-axis we report averaged across 3 seeds episodic reward achieved by the student. Shaded area corresponds to confidence intervals. For a Run task, the confidence intervals are small, so they are not visible. In solid line we report the performance when training using the cross-entropy. In dashed line, we report the performance when training using log probability on the mean action from the expert. All the methods use mean action during evaluation. The black dashed line indicate average (among the same 150 independent evaluations) expert performance for the given expert noise level.

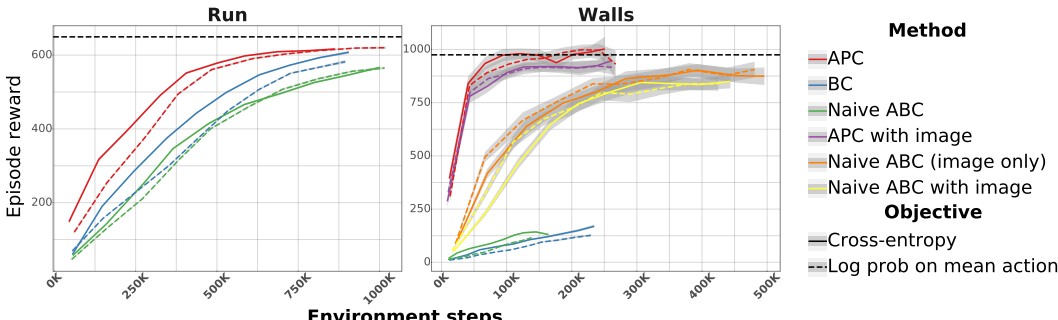

Figure 10: **DAGGER objective sweep with** $\beta = 0.3$. On the X-axis we report the number of environment steps. On the Y-axis we report averaged across 3 seeds episodic reward achieved by the student. Shaded area corresponds to confidence intervals. For a Run task, the confidence intervals are small, so they are not visible. In solid line we report the performance when training using the cross-entropy. In dashed line, we report the performance when training using log probability on the mean action from the expert. All the methods use mean action during evaluation. The black dashed line indicate average (among the same 150 independent evaluations) expert performance for the given expert noise level.