# OpenReview forum: "Data augmentation for efficient learning from parametric experts"
_NeurIPS.cc/2022/Conference — NeurIPS 2022 Accept_

### Official Review · Reviewer_e72c · 2022-07-11

**Rating:** 5
**Confidence:** 5
**Soundness:** 3 good
**Presentation:** 2 fair
**Contribution:** 2 fair

**Summary:**

This paper proposes a simple yet effective way APC, which queries the parametric experts to augment the demonstrations for imitation learning. APC is shown to outperform naive behavior cloning in both expert compression and transferring from privileged expert settings.

**Questions:**

I think the key problem here is that the authors don't explore the methods of querying experts enough. From my point of view, the advantage of APC is totally brought by a new setting where the expert could be queried, which is not an innovation of the methods. I suggest the authors compare APC with more baselines that can query the experts.

Although this method is reasonable, I'm afraid the current technical contribution is far from enough. Adding the Gaussian noise is the simplest way to make a data augmentation, and previous works (e.g., RAD, https://arxiv.org/pdf/2004.14990.pdf) also have systematically studied the data augmentations for general reinforcement learning problems. It could be better if the authors come up with novel methods for data augmentations in your settings.

**Limitations:**

The authors have addressed the limitations.

**Strengths And Weaknesses:**

Pros:
This paper focuses on a novel problem where a parametric expert can be queried. The paper writing is clear and easy to follow. The empirical results do demonstrate the advantage of APC over BC and naive ABC.

Cons:
However, I don't feel this paper proposes any novel idea. One serious issue is that when compared with baselines, all these baselines don't query the expert as APC does. This means that the baselines have fewer demonstrations to imitate from. It is clear that APC will beat naive ABC since APC has more high-quality/ground-truth demonstrations. Therefore, I think this doesn't make enough technical contributions to the problem it studies.

---

> ### Author Response · Authors · 2022-08-02
> **Response to Reviewer e72c**
>
> We would like to thank Reviewer e72c for taking time to review our paper.
>
> Please find below our response to the points from your review.
>
> **APC is shown to outperform naive behavior cloning in both expert compression and transferring from privileged expert settings.**
>
> We want to highlight that we also provide results for kickstarting and DAgger.
>
> **This paper focuses on a novel problem where a parametric expert can be queried**
>
> We want to highlight that this problem is not novel. For example, DAgger is a relatively old approach which also has this assumption.
>
> **I think the key problem here is that the authors don't explore the methods of querying experts enough. From my point of view, the advantage of APC is totally brought by a new setting where the expert could be queried, which is not an innovation of the methods. I suggest the authors compare APC with more baselines that can query the experts.**
>
> We would like to highlight that the setting where experts can be queried is not a new setting (i.e. DAgger). This setting, where experts can be queried, and which was relevant for quite  awhile already, has not yet exploited the method we are proposing, because it is not obvious how to do data augmentation in an expert-driven scenario. In this paper, we explored DAgger and kickstarting settings which were already proposed and studied some time ago. In Figure 5, Figure 6 we report the corresponding results, where by 'BC', we mean the original DAgger and kickstarting methods, and by APC we mean DAgger+APC and kickstarting+APC. While we don't go beyond the studied settings, the innovation of this work is to highlight the importance of producing new actions for perturbed states when doing data augmentation in expert-driven learning.
>
> **Although this method is reasonable, I'm afraid the current technical contribution is far from enough. Adding the Gaussian noise is the simplest way to make a data augmentation, and previous works (e.g., RAD, https://arxiv.org/pdf/2004.14990.pdf) also have systematically studied the data augmentations for general reinforcement learning problems. It could be better if the authors come up with novel methods for data augmentations in your settings.**
>
> Although we agree that adding Gaussian noise is a simple way to augment the data, the point of this paper is to highlight the way the data augmentation should be done in expert-driven learning and that this innovation has not been used, despite prevalence and utility of the expert-driven setting. RAD focuses on the offline RL setting and provides classical well-known data augmentation strategies which only modify the inputs and not the outputs. Our paper argues that it is important not only to augment the inputs but also provide corresponding output actions. Moreover, we show in our work that even with very simple data augmentation strategies, we are able to achieve much higher data efficiency compared to naive approaches which are closer to RAD style.

---

> > ### Comment · Reviewer_e72c · 2022-08-09
> > **Thanks for the response**
> >
> > Thank the authors for the response.
> >
> > I think it is kind of weird to use 'BC' to refer to Dagger, which confuses me about the setting of this paper. I want to thank the authors' response again for clarifying this. I'm willing to increase the score, but I think the issue of not enough baselines is not fully addressed in the response, which is also pointed out by reviewer e72c. Although the offline RL is a different setting, the offline RL algorithms should be easily adapted to the setting of APC, that's why I say the setting in this paper is new which is more like offline RL + queriable expert policy.

---

> > > ### Author Response · Authors · 2022-08-09
> > > **Response to the response**
> > >
> > > We thank you for your response.
> > >
> > > Thank you for this remark. We also agree that it is confusing and we will change it in the final version to make it more clear that it is not 'BC' in DAgger plots, but the real 'DAgger'.
> > >
> > > >  I'm willing to increase the score, but I think the issue of not enough baselines is not fully addressed in the response, which is also pointed out by reviewer e72c. Although the offline RL is a different setting, the offline RL algorithms should be easily adapted to the setting of APC, that's why I say the setting in this paper is new which is more like offline RL + queriable expert policy.
> > >
> > > Generally, BC algorithm works well when the dataset contains very high quality data, which is the case for our setting. Offline-RL algorithms typically provide strategies to deal with the fact that the data might come from a 'bad' policy. It is therefore unclear, whether offline-RL algorithms in our settings would provide a stronger baseline than BC.
> > >
> > > Moreover, in offline RL, it is preferred to be able to learn a Q-function from the data. It is unclear at the moment how APC could be used in this setting and we leave it open for the future work.
> > >
> > > The setting of "offline RL + queriable expert policy" is an interesting one, but it is not as well-defined as the one which we consider - "expert-driven learning with queriable experts". In the setting which you propose, there are some open questions which need to be addressed and practically justified:
> > >
> > > * What should be used as the expert policy? Is it the same policy which generated the data or something else? (note that the data might have been generated from the 'bad' policies)
> > > * What should be the desiderata when the data generating policy is 'bad'?
> > >
> > > We believe that answering these questions is orthogonal to our work and requires its proper investigation.
> > >
> > > Our setting "expert-driven learning with queriable experts" is not novel and it is practically justified as it is encountered in situations which we described in the paper: Kickstarting, DAgger, learning from privileged experts, experts compression. The algorithm like DAgger is a widely used algorithm.
> > >
> > > We appreciate your suggestion on looking into offline RL algorithms and settings. We will add a note in the discussion which clarifies the distinction between our setting and the offline RL, and about possible future steps of using APC idea in these settings.
> > >
> > > We hope that answers your concerns.

---

> > > > ### Comment · Reviewer_e72c · 2022-08-09
> > > > **Thanks for the clarification**
> > > >
> > > > Thank you again for the response, it makes much more sense to me. I will raise the score.

---

### Official Review · Reviewer_immT · 2022-07-11

**Rating:** 6
**Confidence:** 5
**Soundness:** 3 good
**Presentation:** 4 excellent
**Contribution:** 3 good

**Summary:**

They use augmentatio to improve data efficiency with learning from parametric experts. In naive data augmentation approach, parametric experts act same on augmented states and original states. However, APC get different action on augmented states using parametric experts.
They add a samll Gaussian state perturbation to state, and parametric experts predict action on perturbed state.
They experiment on continual control tasks, and show improved data efficiency.

**Questions:**

1. I want to check the change of APC performance according to experts level.
2.  Ablation study of sigma for Gaussian perturbation. (If the perturbatsion is small, a new action sampling from experts can be not needed.)
3. Do you try other perturbation methods on states?

**Limitations:**

In my opinion, their contribution is not enough, because APC can only used with parametric experts. Thus, we have to need well trained experts. Therefore, the scope of utilization is bound to be limited.
Furthermore, baseline is not enough to assure the contribution of APC.

**Strengths And Weaknesses:**

Strength
This paper is well written with proper experiments.
APC significantly improve data efficiency, while naive ABC have almost similar data efficiency with BC.
Their experiement section is well structured for application of their method APC such as Expert compression and Privileged experts.

Weakness
Baselines are not enough. They have to add other offline RL algorithms to compare with APC.
The performance of experts is not revealed in the Figures.

---

> ### Author Response · Authors · 2022-08-02
> **Response to Reviewer immT**
>
> We would like to thank Reviewer immT for taking time to review our paper.
>
> Please find below our response to the points from your review.
>
> **Baselines are not enough. They have to add other offline RL algorithms to compare with APC.**
> Offline-RL is a different setting, where the data is assumed to be coming from some policies, which are not necessarily the expert policies. Moreover, in such scenarios, typically, it is assumed that the policies which produced the data cannot be queried. Therefore, such a scenario is not relevant for our work.
>
> **The performance of experts is not revealed in the Figures.**
>
> We report expert performance via dashed lines in all the figures. We realized that not all the figures have this description. We will amend the description of the figures and add this information.
>
> **Ablation study of sigma for Gaussian perturbation. (If the perturbatsion is small, a new action sampling from experts can be not needed.)**
>
> We provide this ablation in Figure 6 and Figure 7 from Supplementary Material.
>
> **Do you try other perturbation methods on states?**
>
> In this work we only tried Guassian perturbations. However, we want to highlight that the type of perturbation is not a core contribution of this work. It is possible that some structured (non-Gaussian) perturbations could be more efficient and this may be interesting to investigate in future work.
>
> **In my opinion, their contribution is not enough, because APC can only used with parametric experts. Thus, we have to need well trained experts. Therefore, the scope of utilization is bound to be limited. Furthermore, baseline is not enough to assure the contribution of APC.**
>
> As stated in our work, the case of queryable experts arises in multiple important and widely used applications: DAgger, kickstarting, learning from privileged experts, expert compression. For each of these settings, we provide experiments demonstrating the superior performance of our method over the baselines. The baselines which are provided are BC-like (no augmentation) and naive-ABC-like - a straightforward way of using data augmentation in RL, where we would focus on augmenting the inputs but not the outputs. In cases of DAgger, BC-like baseline corresponds to a normal DAgger and in Figure 5, we demonstrated much higher data efficiency of APC.

---

### Official Review · Reviewer_kiE1 · 2022-07-12

**Rating:** 6
**Confidence:** 4
**Soundness:** 3 good
**Presentation:** 3 good
**Contribution:** 3 good

**Summary:**

The paper introduces a data augmentation technique for behavioral cloning algorithms. It consists of adding Gaussian noise to the states used for training and storing the perturbed states and the action the expert policy would produce in such states. Using this augmented dataset, the proposed algorithm achieves better performance than the baselines over three complex continuous tasks.

**Questions:**

While in continuous control environments adding Gaussian noise to a state often produces a perturbed state that could be reached by an agent (i.e. valid states), it is not the case when operating with visual observations. Is there an intuition why data augmentation would be useful in the visual domain?

Assume that, when the expert was trained, it observes in the environment states with mean $\mu$ and standard deviation $\sigma$ (this could be estimated from the last few trajectories during the training of the expert). What is the performance of APC when using zero trajectories and only synthetic data coming from a normal distribution with mean $\mu$ and standard deviation $\sigma$?


Figure 1 and Algorithm 1 appear many pages before the first time they are mentioned in the paper. I suggest putting them when they are mentioned.

**Limitations:**

The authors discuss the possible limitations of this work. The method queries the expert many times, so if expert queries are expensive, their approach might suffer. I suggest providing specific examples for this setting.




---------------------------------------------------------------------------------------------------------------------------------------------------------------------
Edit: updating my score after author-reviewer discussion

**Strengths And Weaknesses:**

The paper is well written and contains many implementation details and ablation studies.

The idea proposed is very simple and effective. Unfortunately, apart from the experimental results in complex environments, the novelty in this work is limited. It is well known that augmenting the dataset in Behavioral Cloning is useful and there are prior works applying observation perturbation and label corrections via experts queries [1,2,3].

The main motivation for using the proposed approach is the case when collecting more data in the environment is expensive, but querying the expert many times is not. The authors motivate this by saying that "an expert policy may be too large to execute due to memory considerations" and that "Many expert-driven learning approaches actually have access to an expert that can be queried", but no specific example is provided to the reader.

The comparison with baselines BC and ABC does not seem very fair. If the policy has enough capacity and the distribution from which we sample the states covers the state-space enough, one could clone the expert by repeatedly querying it on such random states without even seeing a single trajectory. Therefore it is quite straightforward that the proposed method works better than the baselines.

[1] M. Bojarski et al., “End to end learning for self-driving cars,” arXiv preprint, 2016

[2] L. George, T. Buhet, E. Wirbel, G. Le-Gall, and X. Perrotton, “Imitation learning for end to end vehicle longitudinal control with forward camera,” in NeurIPS, 2018.

[3] T. Buhet, E. Wirbel, and X. Perrotton, “Conditional Vehicle Trajectories Prediction in CARLA Urban Environment,” in ICCV, 2019

---

> ### Author Response · Authors · 2022-08-02
> **Response to Reviewer kiE1: Part 1**
>
> We would like to thank Reviewer kiE1 for taking time to review our paper.
>
> Please find below our response to the points from your review.
>
> **It is well known that augmenting the dataset in BC Cloning...via experts queries [1,2,3].**
>
> We thank the reviewer for providing these references. In these references, the authors study very specific IL problems in case of vehicle control and in all the cases, the solution to 'label' corrections actually require quite a lot of prior knowledge (model of the environment) and are hand-crafted. One could consider the proposed solutions as a form of querying the expert, where the expert is a manually constructed policy.
>
> In [1],  the authors use 'shift correction' which requires knowledge of an exact connection between the 'shifted' image and the corresponding 'recovered' label. Moreover, arguably, the added 'shifted' image & 'recovered' label does not correspond to the behavior of the original 'expert'. The authors decided that for a 'shifted' image, the control must behave as 'recovered'. While this could be interpreted as a very specific kind of hand-crafted expert in a very specific domain, the solution strategy proposed in this referenced work is far narrower in scope than the quite general approach we've proposed.
>
> In [2], the 'label' augmentation approach that authors provide corresponds to creating new images via zooms to the obstacles and associating to these images 0 speed.
>
> In [3], the authors consider sequence prediction task and essentially create artificial perturbation in inputs and artificial recovery in the outputs. The authors force that this artificial perturbation leads to a recovery, which is a form of prior knowledge added into the expert policy. Furthermore, generating these perturbations & recoveries requires strong domain knowledge.
>
> In summary, while we appreciate the pointers to this interesting papers that  demonstrate the usefulness of label corrections in specific Imitation Learning scenarios when strong domain knowledge is available [1-3], these approaches are not as useful  for the general Policy Cloning setup where experts can be queried. Moreover, we’ve demonstrated how our simple approach can be applied in a broader range of settings than considered in the above papers. We are happy to include a discussion about these papers in the text.
>
> **The main motivation...but no specific example is provided to the reader.**
>
> In the paper, we do provide examples where access to the 'queryable' expert is assumed:
> *  DAgger & kickstarting - in both cases we assume the ability to query the expert many times, and in both cases we care about data-efficiency when collecting more data. See Section 6, Figure 5 and Figure 6.
> * Expert compression - in this case, we assume that we want to compress a large expert into a smaller one and we study one compression setting. See Section 5.2, Figure 3
> * Learning from experts with privileged information - in this case we assume to have access to an expert which has access to the privileged information. We provide an example of the Insert Peg task where an expert was trained from state features (EASY) and was used to train a student to perform the Insert Peg task from vision (HARD). Using easy-to-train privileged experts to help with learning agents in harder scenarios is a common strategy in RL. See Section 5.3, Figure 4 for more details

---

> > ### Author Response · Authors · 2022-08-02
> > **Response to Reviewer kiE1: Part 2**
> >
> > **The comparison with baselines BC and ABC does not seem very fair...without even seeing a single trajectory. **
> >
> > The comparison to BC makes sense since it is a common strategy of learning from the expert trajectories. Comparing naive ABC makes sense since it is what a straightforward approach of using data augmentation in RL would look like (see [1] for example), i.e., one would perturb the input state and keep the output the same, as it is typically done in supervised learning. That said, it is also important to say that the proposed method APC, for each encountered state, does M (in our experiments, 10) queries to the expert for the perturbed states. Therefore, it requires 10x more queries to the expert compared to the baselines. In the case which you proposed, where one could 'simply' cover the whole state distribution, it would require exponentially more queries to the expert, which may become problematic since it also will require more computation. Our method does smarter state-distribution coverage by only exploiting neighborhood states around the trajectory. Finally, it is quite likely that this 'brute-force' solution may not work as the state-distribution may contain very large amounts of states which are 'uninformative' about the task and therefore lead to a very unbalanced setting.
> >
> > **Therefore it is quite straightforward that the proposed method works better than the baselines.**
> >
> > Our work shows when & how the proposed method can be used in settings where queries of the experts are possible and proposes a non-straightforward way to do data-augmentation. The most straightforward way would be to perturb the inputs but not change the outputs.
> >
> > **While in continuous control environments adding Gaussian noise to a state often produces a perturbed state that could be reached by an agent (i.e. valid states), it is not the case when operating with visual observations. Is there an intuition why data augmentation would be useful in the visual domain?**
> >
> > We only tried to add Gaussian Noise to continuous states. In vision domains, we did try data augmentation techniques in combination with gaussian noise, but we did not produce the corresponding action from the expert with perturbed image observations. In our opinion, using vision perturbations would require an expert which could interpret these perturbations, i.e. training an expert with perturbed images.
> >
> > **What is the performance of APC when using zero trajectories and only synthetic data coming from a normal distribution with mean μ and standard deviation σ ?**
> >
> > Even though this is a reasonable way of thinking about the motivation for our setting, this concrete approach will generally fail when state-action spaces are not very small.  Indeed, our APC approach could be interpreted as performing the proposed sampling in the vicinity of "good" trajectories.  While we don’t believe it is worthwhile to include the proposed very weak baseline throughout the paper, we note that in our experiments, adding more trajectories generally helps for all the methods. This suggests that simple Gaussian sampling would not sufficiently capture the relevant part of the space. Finally, one could consider an alternative approach of fitting a state model on the dataset and then sampling from it and querying the expert. We consider such an approach complicated enough such that it deserves a separate proper investigation.
> >
> > **Figure 1 and Algorithm 1 appear many pages before the first time they are mentioned in the paper. I suggest putting them when they are mentioned.**
> >
> > Thank you, we will rearrange the figures in the paper.

---

> > > ### Comment · Reviewer_kiE1 · 2022-08-07
> > > **Response to authors**
> > >
> > > I thank the authors for their response.
> > >
> > > > In the paper, we do provide examples where access to the 'queryable' expert is assumed:
> > >
> > > I suggest the authors to add to the main paper more real-world practical examples. I appreciate the examples provided, but since the main motivation for this approach comes from this particular setting, I would like to see more examples where querying the expert is not expensive.
> > >
> > > > The comparison to BC makes sense since it is a common strategy of learning from the expert trajectories. Comparing naive ABC makes sense since it is what a straightforward approach of using data augmentation in RL would look like (see [1] for example), i.e., one would perturb the input state and keep the output the same, as it is typically done in supervised learning. That said, it is also important to say that the proposed method APC, for each encountered state, does M (in our experiments, 10) queries to the expert for the perturbed states. Therefore, it requires 10x more queries to the expert compared to the baselines.
> > >
> > > After reading the authors' answers, I am more convinced that it is fair to compare the proposed method against DAGGER. I have to say that it is quite odd at first sight to have a method that can query the expert many more times than the baseline and it seems obvious that such a method would outperform the baseline. However, the fact that the method is able to query the expert more times and in the right way is actually the strength of the proposed method.
> > >
> > > > Even though this is a reasonable way of thinking about the motivation for our setting, this concrete approach will generally fail when state-action spaces are not very small. Indeed, our APC approach could be interpreted as performing the proposed sampling in the vicinity of "good" trajectories. While we don’t believe it is worthwhile to include the proposed very weak baseline throughout the paper, we note that in our experiments, adding more trajectories generally helps for all the methods. This suggests that simple Gaussian sampling would not sufficiently capture the relevant part of the space. Finally, one could consider an alternative approach of fitting a state model on the dataset and then sampling from it and querying the expert. We consider such an approach complicated enough such that it deserves a separate proper investigation.
> > >
> > > One could model the state distribution encountered by the expert using a multimodal distribution and then sample from it without observing new data. The approach would not necessarily fail, since the expert would be queried in the states visited by the expert, so at most one could recover the expert's performance even before acting in the environment. This could be used together with APC to further increase sample efficiency. I agree though that this would require a separate investigation.
> > >
> > > I will update the scores for the submission at the end of the discussion period.

---

> > > > ### Author Response · Authors · 2022-08-09
> > > > **Response to a response**
> > > >
> > > > We thank you for your response.
> > > >
> > > > > I suggest the authors to add to the main paper more real-world practical examples. I appreciate the examples provided, but since the main motivation for this approach comes from this particular setting, I would like to see more examples where querying the expert is not expensive.
> > > >
> > > > We appreciate your suggestion. We are open to add more examples in the discussion section on top of the ones we provided, especially if there is anything specific.
> > > >
> > > > DAgger and its derivates are widely used in practice algorithms. In DAgger, the main motivation is to reduce the number of executions / interactions. It is implicitly assumed that querying the expert is not expensive. Therefore, any setting where executions / interactions are expensive but querying is cheap, will be suitable for this method. Some examples are [1-2].
> > > >
> > > > The setting which corresponds to learning from privileged experts is also a very relevant in practice setting. It is often much easier to train a policy which has additional information in the inputs (like well-constructed features), but such policy might not be good enough for more general setting (it is hard to construct good features). A typical strategy is to use this privileged policy to help learning a more general policy, which receives less information to the input, such as vision instead of features. This approach can be used in robotics tasks.
> > > >
> > > > The settings which we studied in the paper can be used in many applied problems. We will add a few more examples of applied problems where these settings are relevant in the discussion.
> > > >
> > > > > After reading the authors' answers, I am more convinced that it is fair to compare the proposed method against DAGGER. I have to say that it is quite odd at first sight to have a method that can query the expert many more times than the baseline and it seems obvious that such a method would outperform the baseline. However, the fact that the method is able to query the expert more times and in the right way is actually the strength of the proposed method.
> > > >
> > > > We agree with your observation, thank you.
> > > >
> > > > > One could model the state distribution encountered by the expert using a multimodal distribution and then sample from it without observing new data. The approach would not necessarily fail, since the expert would be queried in the states visited by the expert, so at most one could recover the expert's performance even before acting in the environment. This could be used together with APC to further increase sample efficiency. I agree though that this would require a separate investigation.
> > > >
> > > > This could be a reasonable approach, but as we and you noted, it is complicated enough so it requires a separate investigation.
> > > >
> > > > **References:**
> > > >
> > > > [1] Neural probabilistic motor primitives, Josh Merel , Leonard Hasenclever , Alexandre Galashov, Arun Ahuja, Vu Pham, Greg Wayne, Yee Whye Teh, & Nicolas Heess
> > > >
> > > > [2] Information asymmetry in KL-regularized RL, Alexandre Galashov, Siddhant M. Jayakumar, Leonard Hasenclever, Dhruva Tirumala, Jonathan Schwarz, Guillaume Desjardins, Wojciech M. Czarnecki, Yee Whye Teh, Razvan Pascanu, Nicolas Heess

---

### Official Review · Reviewer_pnsx · 2022-07-13

**Rating:** 6
**Confidence:** 4
**Soundness:** 3 good
**Presentation:** 3 good
**Contribution:** 3 good

**Summary:**

This paper proposes an approach for imitation learning from parametric experts, by augmenting the data and then querying the experts for action labels on the new states. The authors emphasize two practical settings where learning from parametric experts is useful - expert compression and learning from privileged experts. Includes evaluation on continuous control mujoco tasks (humanoid and peg insertion).


**Questions:**

-> How does the proposed approach compare to regular dagger.  Specifically, how much worse is the policy performance (if at all), if the policy isn’t rolled out to collect new states, but instead the recorded states were just perturbed to get new states, and the expert was then queried for new actions ? What are the sample efficiency gains ?

-> What is the performance of the expert MPO policies on the domains reported?

-> For expert compression results, please plot ratio of performance of a low capacity model to that of the high capacity model for each method for easier comparison of compression ability.


**Limitations:**

The limitation are adequately discussed, primarily being requiring a parametric expert.

**Strengths And Weaknesses:**

Originality -

-> The proposed idea is very simple and intuitive. Data augmentation is commonly used for machine learning approaches and recently also in RL for improved performance [1,2]. Given parametric experts, querying them for action labels on the new augmented states seems like it would give rise to better performance, since more information is being extracted from the expert. This is similar to how in dagger when new states are encountered, the expert is queried for labels, except here the new states don’t require a policy rollout but rather a direct perturbation of the recorded states. While the paper does include experiments on dagger with data augmentation, I’d be curious to see how regular dagger compares to the proposed approach, in terms of policy performance and overall sample efficiency.

-> The paper is missing a related works section that goes into depth about prior work in the area, but the structure of the paper does include some discussion on prominent imitation learning methods (dagger, kickstarting, privileged experts) but doesn’t mention specific works adequately.

[1] : Reinforcement Learning with Augmented Data (Lashkin et al.)
[2] : Regularizing Deep Reinforcement Learning from Pixels (Kostrikov et al.)

Significance -

-> The proposed method can be of interest to a large section of the community interested in imitation learning, and is quite simple to adopt and implement. Sample efficiency is critical especially for robot learning, and the proposed approach shows the ability to learn control policies in the very low data regime. The only caveat is that the method needs parametric experts, but the authors are very clear about this and point out practical areas where the proposed approach is relevant - i.e expert compression and learning from privileged experts. The latter is specifically relevant for robotics, since sim to real pipelines [3,4] can adopt this idea and become more efficient.

[3] : Rapid Motor Adaptation (Kumar et al.)
[4]: Solving Rubik’s Cube with a Robot Hand (OpenAI)

Quality -

-> The experimental evaluation seems quite thorough (performance is reported on 150 random environment instantiations). The authors show that the proposed approach fits the expert data much better than the baselines given an offline dataset, and the results on learning from privileged experts do seem to show that the proposed approach is effective

-> However, on the expert compression task, the only environment where there’s a clear advantage for the proposed method is humanoid run. On peg insertion, the baseline’s high capacity model has quite poor performance so it’s difficult to judge compression ability, and on humanoid walls the relative compression of the proposed approach and the baseline appear similar. I would recommend defining a new metric for compression (i.e ratio of performance of a low capacity model to that of the high capacity model) so that performance can be more accurately judged.

Clarity -

The paper is well written and motivated, and easy to follow. It includes clear algorithm boxes and equations that are sufficient to understand the details of the method.

---

> ### Author Response · Authors · 2022-08-02
> **Response to Reviewer pnsx**
>
> We would like to thank Reviewer pnsx for taking time to review our paper.
>
> Please see our response to the points from your review.
>
> **How does the proposed approach compare to regular dagger. Specifically, how much worse is the policy performance (if at all), if the policy isn’t rolled out to collect new states, but instead the recorded states were just perturbed to get new states, and the expert was then queried for new actions ? What are the sample efficiency gains ?**
>
> We believe that there is a misunderstanding of the DAgger section from the paper. In this section, we actually provide a comparison between "regular" DAgger (referred to as BC in the Figure 5) against DAgger which is augmented with APC (APC in Figure 5). In case of 'BC', the expert is queried only on the states which were encountered during the student/expert unrolls.
> We realized that Figure 5 is confusing when 'BC' is used instead of 'DAgger'. We will replace this name and amend the description below the figure as well as the DAgger section.
>
> **What is the performance of the expert MPO policies on the domains reported?**
>
> For all the plots (except Figure 3), we report expert performance on the corresponding domain using dashed lines. We realized that not all the figures describe that dashed lines are corresponding expert performance and we will amend this.
>
> **On peg insertion, the baseline’s high capacity model has quite poor performance so it’s difficult to judge compression ability, and on humanoid walls the relative compression of the proposed approach and the baseline appear similar.**
>
> In fact, the performance on insert peg is quite poor for BC and we generally found it very challenging to train a policy on this task. The reason for is due to the fact that in Insert Peg, the rewarding state corresponds to a situation where arm inserted a sword into a hole and does not move (and episode is not finished until 1000 time steps had elapsed), therefore the dataset consists of very similar, not-diverse data. In supplementary material (due to space constraints), we conducted an experiment with what we called 'short trajectories', where a dataset consisted of 1 full trajectory (1000 timesteps or more) and N-1 (where N is the number of trajectories) short trajectories containing only 200 timesteps each. This experiment is motivated by a Robotics setting where a robot may fail too early in the episode and may succeed only a few times. Therefore, the corresponding dataset would contain a lot of short trajectories. For more information see Section 6 from Supplementary Material and Figure 1 from Supplementary Material. In this experiment, we noticed that BC performs well on insert peg when only short trajectories are used (see Figure 1 from Supplementary material). As mentioned above, for insert peg, there will be a lot of variations in state distribution around the beginning of the episode and less variations afterwards. APC is able to deal well with both situations, whereas BC only works when the dataset contains short trajectories.
>
> **For expert compression results, please plot ratio of performance of a low capacity model to that of the high capacity model for each method for easier comparison of compression ability.**
>
> This is a very good suggestion. As a way to address it, we propose to add additional information in the legend, where alongside the student torso architecture, we will also add a ratio of the compressed student capacity versus the highest capacity model.
>
> **The paper is missing a related works section that goes into depth about prior work in the area, but the structure of the paper does include some discussion on prominent imitation learning methods (dagger, kickstarting, privileged experts) but doesn’t mention specific works adequately.**
>
> The paper is structured such that it includes Related work discussion in Introduction. We thank you for highlighting this point and we will add more discussion on related work in the area in the introduction.

---

> > ### Comment · Reviewer_pnsx · 2022-08-10
> > **Rebuttal Response**
> >
> > I thank the authors for their response and clarifications. The author clarifications about the dagger/BC naming in the paper as well as the peg insertion experiments on shorter trajectories in the supplemental section serve to strengthen the paper, but not sufficiently to increase my score since I'm already in favor of acceptance.

---

### Author Response · Authors · 2022-08-02
**Thank you**

We thank all the reviewers for taking the time to read our paper and provide the review.

Please find specific answers in the corresponding sections.

---

### Meta-Review · Area_Chair_bGHt · 2022-08-25

**Recommendation:** Accept
**Confidence:** Certain

**Metareview:**

This paper studies an interesting problem, and overall the reviewers agreed the exposition and validation are sufficient. We encourage the authors to consider the issues raised by the reviewers and further improve the work in the final version.


**Award:**

No

---

### Decision · Program_Chairs · 2022-09-14

Accept